Corrected: Publisher correction

# Mammalian display screening of diverse cystine-dense peptides for difficult to drug targets

Zachary R. Crook [1], Gregory P. Sevilla[1], Della Friend[2], Mi-Youn Brusniak [1], Ashok D. Bandaranayake[1],
Midori Clarke [1], Mesfin Gewe[2], Andrew J. Mhyre [1], David Baker[3], Roland K. Strong[2],
Philip Bradley[4] & James M. Olson[1]

Protein:protein interactions are among the most difficult to treat molecular mechanisms of disease pathology. Cystine-dense peptides have the potential to disrupt such interactions, and are used in drug-like roles by every clade of life, but their study has been hampered by a reputation for being difficult to produce, owing to their complex disulfide connectivity. Here we describe a platform for identifying target-binding cystine-dense peptides using mammalian surface display, capable of interrogating high quality and diverse scaffold libraries with verifiable folding and stability. We demonstrate the platform's capabilities by identifying a cystine-dense peptide capable of inhibiting the YAP:TEAD interaction at the heart of the oncogenic Hippo pathway, and possessing the potency and stability necessary for consideration as a drug development candidate. This platform provides the opportunity to screen cystine-dense peptides with drug-like qualities against targets that are implicated for the treatment of diseases, but are poorly suited for conventional approaches.

[1] Clinical Research Division, Fred Hutchinson Cancer Research Center, 1100 Fairview Avenue N Room D4-100, Seattle, WA 98109, USA. [2] Basic Sciences, Fred Hutchinson Cancer Research Center, 1100 Fairview Avenue N Room B3-183, Seattle, WA 98109, USA. [3] Department of Biochemistry, University of Washington, Molecular Engineering and Sciences, Box 351655, Seattle, WA 98195, USA. [4] Public Health Sciences, Fred Hutchinson Cancer Research Center, 1100 Fairview Avenue N Room M1-B514, Seattle, WA 98109, USA. Correspondence and requests for materials should be addressed to P.B. (email: pbradley@fredhutch.org) or to J.M.O. (email: jimmyo@uw.edu)

In identifying targets for drug discovery efforts, numerous proteins have emerged that have proven impossible or impractical to inhibit. Examples include most proteins at the core of neurodegenerative disease, such as Aβ, tau, or huntingtin[1], as well as long-known cancer mediators like c-Myc[2], KRas[3], and TEAD[4]. TEAD is at the core of the oncogenic Hippo pathway, which plays a critical role in wound repair and contact inhibition[5], and is commonly dysregulated in many human cancers, including liver, breast, colon, lung, prostate, and brain[6–11]. The signaling pathway culminates in the intranuclear interaction of TEAD, a transcription factor, and its transcriptional co-activator YAP (or TAZ)[12,13]. This is exemplary of an "undruggable" target, most of which have pathological activities reliant on protein: protein interactions. Conventional screening campaigns with small molecule libraries have had difficulty identifying specific, high-affinity binders capable of disrupting protein–protein interactions[4,14–19]. Meanwhile, antibodies are capable of disrupting protein:protein interactions, but they have trouble accessing the core of solid tumors[20] and targets in the cytosol.

Drug-like, cystine-dense peptides (CDPs) of approximately 10–80 residues occupy a unique mid-sized medicinal space. They are not only capable of interfering with protein:protein interactions, but are small enough to access compartments beyond the reach of antibodies. Found throughout the evolutionary tree, native CDPs with drug-like roles include protease inhibitors[21], venom ion channel modulators[22], and peptide antimicrobials[23]. The calcine knottins are also notable, as they access and retain function in the cytosol (despite its reducing environment) to activate sarcoplasmic reticulum-resident ryanodine receptors[24,25]. Beneficial pharmacologic properties of drug-like CDPs can be attributed to a series of intra-chain disulfide crosslinks that stabilize the peptides, improve binding properties by limiting flexibility of the binding interface, and render many of them resistant to proteases, which reduces immunogenicity[26]. Despite this, there are only a handful of CDPs in the clinic or in trials (e.g., linaclotide, ziconotide, ecallantide, and tozuleristide), a dearth that we attribute to insufficient screening efforts for novel agents.

Screening for a target-engaging protein is a well-established practice, with some promising work using drug-like CDP scaffolds[27–30]. However, these screens have been limited to the handful of discrete native scaffolds that are known to fold into a single disulfide-driven tertiary structure, typically varying only one face or loop to create diversity[27,31]. A diverse CDP library, using millions of variants from thousands of different scaffolds, represents an opportunity to exploit native conformational diversity while maintaining their beneficial drug-like properties. To this end, we developed a mammalian surface display platform optimized for the folding of CDPs, validating it on a highly diverse library of thousands of native CDPs by using both high-throughput mammalian display screening and HPLC to evaluate their expression and stability. Furthermore, we demonstrated its capabilities in rational peptide design screening by identifying a computationally designed CDP that disrupts the YAP:TEAD dimer. This peptide was further optimized for sub-nanomolar equilibrium dissociation constant ($K_D$), and demonstrated the protease resistance, reduction resistance, and thermostability of a promising CDP therapeutic candidate. By leveraging this platform, diverse drug-like peptide libraries can be used to identify therapeutic candidates for difficult-to-drug targets.

## Results

### Choice and validation of mammalian display for CDP screening. 
*E. coli* and *S. cerevisiae* are routinely used for surface display screens to find target-binding peptides (yeast have the advantage of the eukaryotic secretory pathway's oxidative environment to aid disulfide formation)[32,33], yet the variety of CDP scaffolds being reliably surface displayed or secreted is limited[27]. Both species natively secrete fewer than 50 proteins with cysteine-rich domains, compared to the human secretome, of which over 1400 genes (~20%) contain such domains (Supplementary Table 1). Therefore, while bacteria and yeast display are effective systems for many specific, vetted scaffolds, mammalian cells were attractive for diverse, poorly-characterized library screening because they routinely secrete a wide variety of proteins with cysteine-rich segments.

We used a modified version of the Daedalus vector[34] to express peptides tethered to suspension-adapted 293 Freestyle (293F) cells (Figs. 1a, b), with a scaffold based on the Type II transmembrane protein FasL. The vector, named SDGF (Surface Display GFP FasL) (Supplementary Fig. 1), confers specific labeling of cells expressing ligands for target proteins (Fig. 1c), and a single transduction event induces sufficient CDP expression on their surface to become clearly stained by fluorescent binding partners, allowing for efficient enrichment screening (Fig. 1d).

### Diverse native CDPs fold properly in mammalian display. 
For the platform to be useful, CDPs must be displayed as a well-folded species, which we assessed by measuring surface expression and protease resistance, both of which correlate with protein stability[35,36]. To test this, we created a library of 10,000 native cystine-dense peptides or protein fragments, representing diverse taxonomic groups (Fig. 2a). Oligonucleotides encoding these peptides were synthesized as a pool, and cloned into the surface display vector. For these experiments, we used a variant of SDGF, called SDPR (Surface Display Protease Resistance) containing a C-terminal 6xHis tag and mutating all surface-exposed trypsin (basic) and chymotrypsin (aromatic) sensitive residues (Supplementary Fig. 1). This library includes well-characterized drug-like CDPs (e.g., knottins and defensins) but is largely made of cysteine-rich fragments of larger, structurally uncharacterized proteins. Note that, while some library members are unannotated and may not be natively secreted, we still use the term cystine-dense peptide, as their secretion by the mammalian cell creates a permissive environment for cystine formation.

After a control treatment or limited trypsinization, followed by dithiothreitol (DTT) reduction (to release 6xHis tags from proteolysed peptides), cells were stained with iFluor 647-labeled anti-6xHis antibody to quantitate remaining intact surface peptides. Trypsin-treated and untreated cells were sorted into one of four populations by surface stain fluorescence, with each peptide's distribution between the four populations (determined by high throughput sequencing) facilitating measurement of surface protein levels in either condition (Supplementary Fig. 2a). This is because, in the case of a well-expressed peptide, cells expressing it would stain brighter, being preferentially distributed into the high fluorescence populations. This manifests as enrichment in the higher staining populations relative to the lower stained populations. The relative distribution of each peptide within the populations is incorporated with each population's cell count and median fluorescence, yielding a unitless number corresponding to the average fluorescence of a cell expressing that peptide. A similar technique using yeast display was recently validated for designed, cysteine-free peptides[37], but such an analysis for CDPs cannot be performed in conventional yeast display, as the Aga1/2 scaffold is held together with disulfide bonds. This high-throughput, quantitative protein content assay allows us to identify well-folded CDPs by those that confer strong surface staining to cells (high content) and/or retain their staining after protease treatment (protease

resistant). From this analysis, many CDPs from this diverse library (729 of 4298 that passed quantitation thresholds) appear well-folded on the cell surface (Fig. 2b; high content/protease resistant peptides are defined as those residing above the diagonal line).

A CDP that expresses well and/or is resistant to protease may be well-folded in the context of tethering to the mammalian cell surface, but this would be of limited therapeutic relevance if surface folding failed to translate into behavior as a soluble product. To see whether surface folding correlates with drug-like peptide characteristics, 604 library members were produced in small scale as secreted peptides. This group is enriched for known drug-like CDPs, and 41% are well-folded in surface display. A peptide's mobility by reversed-phase HPLC (hereafter referred to as HPLC) is influenced by its structure, so we define a well-folded soluble peptide as one that presents 1–2 peaks (one dominant peak with 0 or 1 minor peaks) both before and after reduction (10 mM DTT). Altered mobility after reduction demonstrates disulfide-driven tertiary structure, though a lack of mobility change could be evidence of either no disulfides, or resistance to reduction. In all, 45% of the tested peptides are well-folded

as soluble peptides. However, properly folded soluble peptides (1-2 peaks) are more often found to fold well on the surface (high content / protease resistant), while peptides that fold poorly (3+ peaks) or fail to secrete (0 peaks) are more likely to demonstrate poor surface folding (low content / protease sensitive) (Figs. 2b–d and Supplementary Table 2). This significant correlation ($P < 1 \times 10^{-6}$ by concordance of surface and soluble folding properties before vs. after HPLC classifier shuffling) was seen for knottins and defensins ($N = 454$), which are well-characterized drug-like CDPs with predictable folding patterns, as well as for other, less structurally characterized peptides ($N = 150$), suggesting a wealth of drug-like CDPs exists in nature beyond well-defined, annotated examples.

Our surface content quantitation assay validated well, but conventional sequence enrichment analysis can also be applied to the dataset. If there is indeed a high correlation between surface content (one of the two measures of surface folding, along with protease resistance) and proper folding as a soluble peptide, we would expect to see enrichment of 1–2 Peak peptides in the sorted cells with higher surface staining. Analyzing the four sorted populations of varying fluorescence ranges (lowest, low, high, and

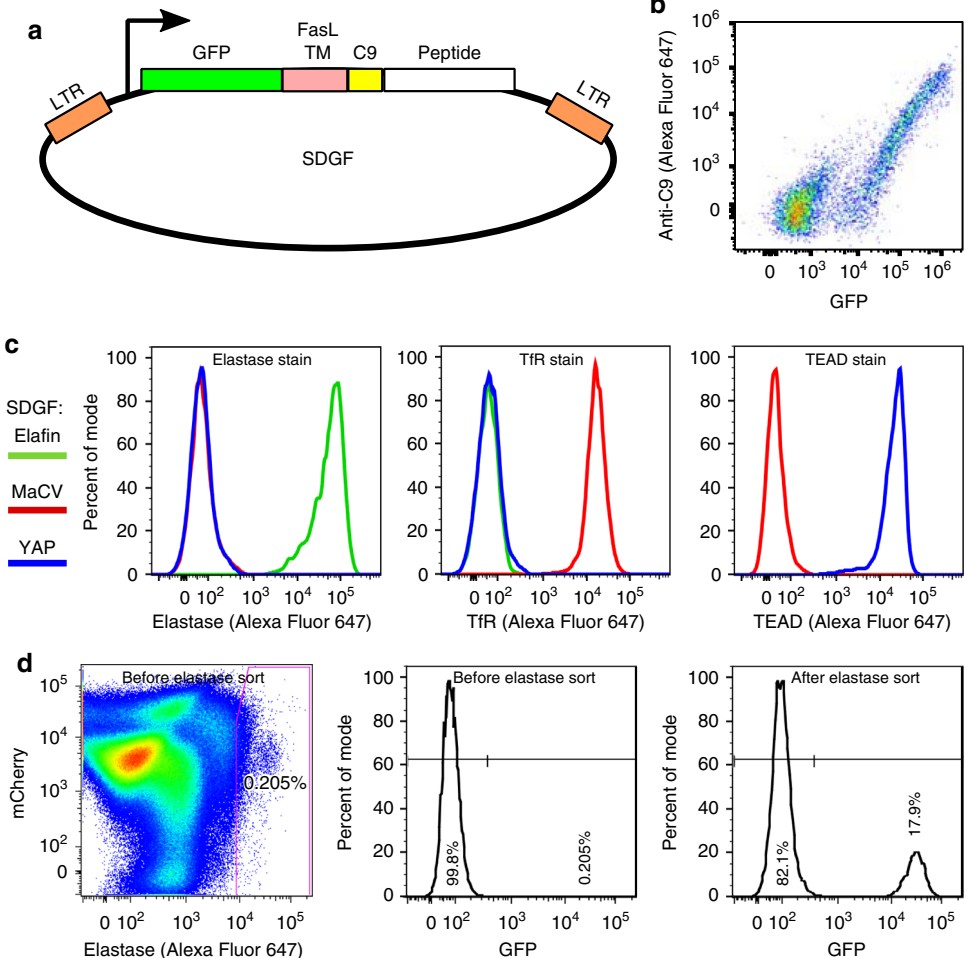

**Fig. 1** Design and validation of mammalian surface display vector SDGF. **a** Illustration of the lentivector SDGF. A variant with mCherry substituting GFP was also constructed, called SDRF. **b** Cells transfected with SDGF and stained with Alexa Fluor 647 labeled anti-C9 antibody. **c** Cells expressing elafin, Machupo virus glycoprotein (MaCV), or YAP via SDGF were stained with Alexa Fluor 647 labeled elastase (left), or with Alexa Fluor 647 labeled streptavidin plus either biotinylated transferrin receptor ectodomain (TfR, middle) or biotinylated YAP-binding domain of TEAD (right). **d** Cells expressing SDGF-elafin were mixed with cells expressing SDRF-MaCV at approximately a 1:500 dilution. The cells were stained with Alexa Fluor 647 labeled elastase, and were flow sorted by Alexa Fluor 647 content (left) with a generous gate to collect all labeled cells. The resulting proportion of green (SDGF) cells increased from 0.2% pre-sort (middle) to 17.9% post-sort (right)

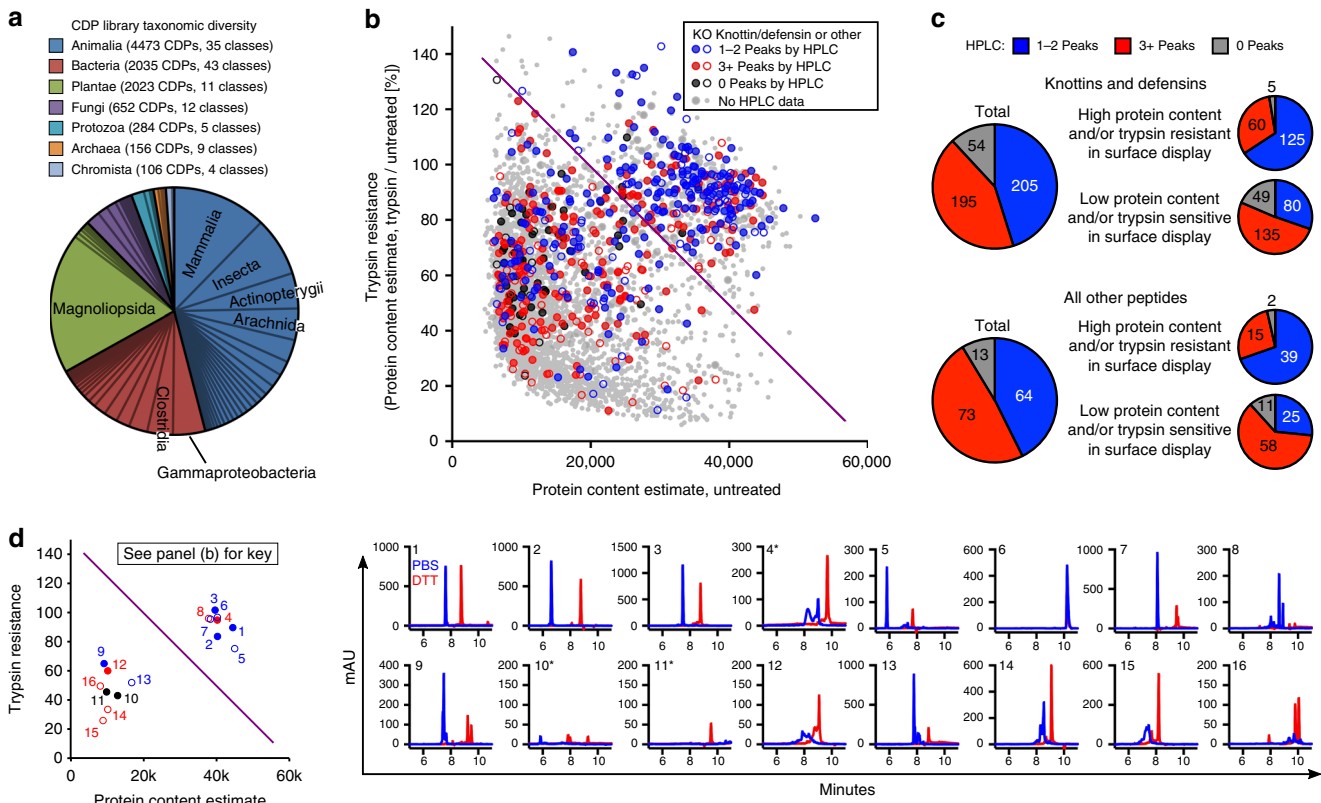

**Fig. 2** Cell surface folding of displayed cystine-dense peptides correlates with proper folding as soluble peptides. **a** Taxonomic diversity of the library. Classes with > 300 library members are named. 270 peptides had missing or corrupt taxonomic lineages. **b** Of 9999 CDPs cloned into SDGF and transduced as a pool, 4298 passed read abundance thresholds and were quantitated for surface protein content in untreated cells (X axis) and trypsin-treated cells (Y axis, as a percentage of untreated surface protein content to measure trypsin resistance). The diagonal line defines an arbitrary cutoff between "High protein content and/or trypsin resistant" (HC/TR) and "Low protein content and/or trypsin sensitive" (LC/TS). Symbols were assigned by HPLC validation performance (1-2 Peaks, blue; 3+ Peaks, red; 0 Peaks, black; or no HPLC data, gray) and by "Knottin/Defensin" (filled) or "Other" (empty, or small gray) classification. Values represent the average of two independent replicates of the complete experiment. **c** Secreted protein HPLC performance, as categorized by peptide classification and by protein content / trypsin resistance. For both peptide categories, the correlation between surface folding (HC/TR vs LC/TS) and HPLC performance (1-2 Peaks vs 3+ or 0 Peaks) was highly significant ($P < 1 \times 10^{-6}$ by concordance of surface and soluble folding properties before vs. after HPLC classifier shuffling). **d** HPLC traces of 16 representative peptides under native (PBS, blue) or reducing (DTT, red) conditions. Peptides 4, 10, and 11 contain potential glycosites (NXS/T) and are marked with an asterisk. Plot on the left is same as in **b**, with only the 16 representative peptides displayed, matched by number to the HPLC plots. Full protein information for the 16 representative peptides is found in Supplementary Table 2. See Supplementary Fig. 2, the Methods, and the Supplementary Methods for detailed illustrations and protocols for high throughput surface protein content quantitation

highest), peptides were ranked within each population by their enrichment or depletion vs. input. Using un-weighted gene set enrichment analysis[38], we indeed see that 1−2 Peak peptides cluster among the most-enriched genes in the two high-staining sorted populations. Conversely, 1−2 Peak peptides are depleted in the low-staining populations (Supplementary Fig. 2b). This confirms the correlation between surface expressed and secreted CDP folding behavior.

**Protein context and glycosylation affect displayed CDPs.** Only 17% of the diverse test library folds well on the surface, which could be related to the fact that most of the library is made of cystine-dense fragments of larger proteins. These fragments may be natively unstructured or have context-dependent structure. After parsing the library by the fraction of the full, native protein occupied by the displayed peptide, peptides that make up ≥ 50% of their full protein sequence (e.g., a knottin peptide and its signal sequence) appear well-folded by surface display 40% of the time (Fig. 3a). This is reduced to 25 and 12% for peptides occupying 25−50 and < 25% (respectively) of their total protein size, the latter category representing 70% of the total library. This supports

the theory that CDP folding is often context-dependent. However, the correlation between surface display folding and soluble peptide folding is independent of native protein context.

Proteins secreted from mammalian cells often have favorable glycosylation profiles for stability and reduced immunogenicity of biologics[39]. However, glycosylation may prove problematic for drug-like peptides, as it could alter their size and protein product homogeneity. Investigating N-linked glycosites (NXS/T) in the test library indeed shows that CDPs that fold properly by surface display, but that also have a glycosite, are significantly (P = 0.0191 by two-tailed Chi Square test) less likely to demonstrate 1−2 clean HPLC peaks when secreted (Fig. 3b). However, glycosites seem to improve surface expression, as glycosite-containing CDPs are more likely to appear properly folded at the cell surface than those without glycosites (24 vs 16%; P < 0.0001 by two-tailed Chi Square test), a result that is unrelated to protein context (Supplementary Table 3). This supports the notion that glycosylation aids protein folding and/or stability, but may compromise product homogeneity. The characteristics that influence surface folding, including protein context, glycosylation, and others not elaborated on in this

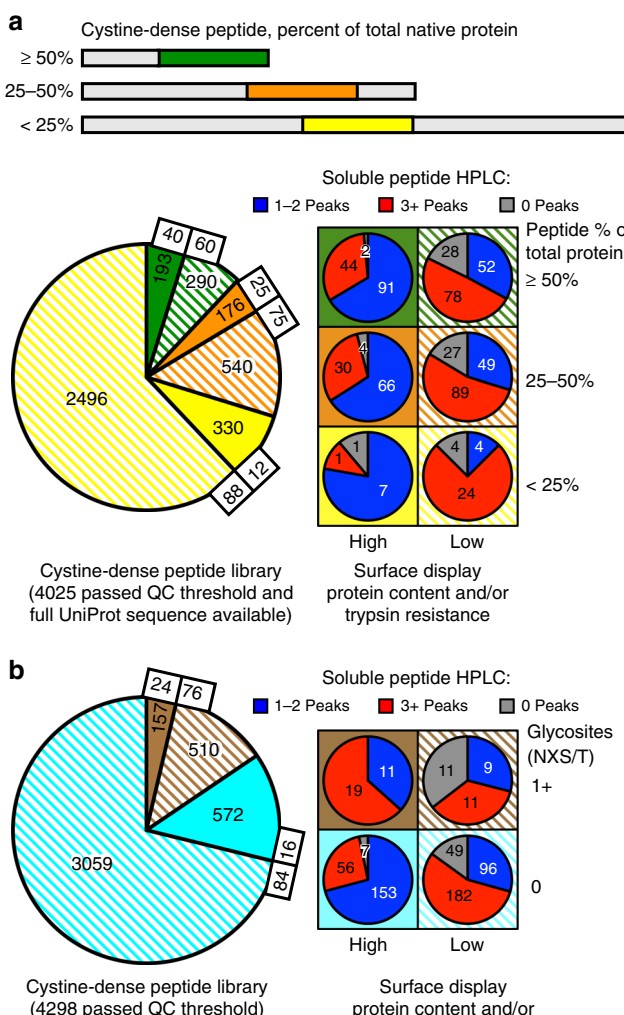

**Fig. 3** Effects of a CDP's native protein context, and glycosylation, on surface and solution properties. **a** Categorizing the tested CDPs by the proportion of the native, full-length protein they represent, either < 25% (yellow), 25-50% (orange), or ≥ 50% (green). The measured surface display protein content/trypsin resistance properties (solid, high; dashed, low) and HPLC classification (1-2 Peaks, blue; 3+ Peaks, red; 0 Peaks, gray) are shown per category. Proportions of high content / trypsin resistant peptides are significantly different ($P < 0.0001$) between all categories. **b** Same analysis as in **a**, but categorized by the presence (brown) or absence (cyan) of canonical N-linked glycosites (NXS or NXT) within the peptide. The presence of a glycosite correlates with a significantly ($P < 0.0001$) higher probability of having high surface display protein content / trypsin resistance, but such high protein content / trypsin resistant, glycosite-containing peptides (solid brown) have significantly fewer ($P < 0.0001$) 1–2 Peak peptides than expected when compared to high protein content/trypsin resistant CDPs that lack NXS or NXT sites (solid cyan). Significance calculated by two-tailed Chi Square test

manuscript (e.g., organism of origin, cysteine topology, and amino acid content), are shaping future library construction that preserves diversity but avoids CDPs with a high likelihood of misfolding. Combined with mutagenesis methods (e.g., error-prone PCR), screens using CDP libraries with a diversity of > $10^7$ variants are well within the platform's capabilities.

**Mammalian CDP screening to identify TEAD-binding optides**. CDP expression in mammalian display facilitates high diversity

screening to find drug-like peptides that interact with a target of interest. Much like antibody generation, the nature of this interaction can be evaluated through secondary assays to determine potential therapeutic utility, as an interaction is not assumed to alter activity. However, many targets are well-characterized structurally, which offers the possibility of using rational design methods to produce candidates that would not simply interact with the target, but would do so in a way predicted to alter its activity in a relevant fashion. Rosetta protein design methods are particularly amenable to interactions driven by well-ordered secondary structure elements[40,41], such as the aforementioned YAP:TEAD interaction. Peptides that target this interaction, based on YAP itself, have been tested[42], but they lacked potency and the demonstrable stability of CDPs, calling into question clinical utility.

We therefore sought out to generate a TEAD-binding CDP that would interrupt YAP:TEAD dimerization, which could inhibit its function. We use the term "optide" (optimized peptide) to describe any CDP, native or designed, that has been further optimized by mutation or chemical alteration for beneficial pharmacologic properties. Because the YAP:TEAD interaction is structurally well-characterized[43], we used a Rosetta protein design approach to design optides capable of binding to TEAD at any of the three characterized YAP binding interfaces, basing the protein design scripts on this published structure. The Methods contain a detailed description of the Rosetta methodology. In brief, small fragments of YAP from the published YAP:TEAD co-crystal structure were tested for compatible engraftment onto CDP scaffolds, which are then tested for steric hindrance at the TEAD interface (i.e. overlap with TEAD in structural space) where the YAP graft was found. Engrafted scaffolds modeled to be free of steric hindrance went through a round of design to introduce residues predicted to strengthen the interaction. The end product is a modeled optide:TEAD interaction, with 7533 such models generated for testing.

We will note that the scaffolds were de novo designed, based on α-helix rich structures with predicted thermostability and further stabilized by the introduction of cysteines at locations compatible for cystine formation. The library contained peptides with 6 cysteines, and of similar size (30–41 amino acids) to the native CDPs that were validated for surface stability. However, scaffolds were not based on native CDPs. This is because most well-characterized drug-like CDPs, such as knottins and defensins, contain structures that are rich in loops[44]. Such peptides may indeed have drug-like properties, but from a design perspective, Rosetta is optimized for secondary structure-driven interactions[40,41], so we predicted that our chances of success at identifying a rationally designed TEAD inhibitor would be greater with a CDP library dominated by α-helices, rather than loops.

The designed optides were cloned as a pool into SDGF. After transduction and expression in 293F cells, the library was screened for binding with biotinylated TEAD (200 nM YAP-binding domain of TEAD with 200 nM Alexa Fluor 647-labeled streptavidin) and expanded over four rounds of sorting (Figs. 4a, b). Hits were tested as singletons for TEAD binding and counter-screened for non-specific streptavidin binding. Two hits, referred to as TB1G1 and TB2G1, targeted Interface 2 and showed strong enough TEAD binding to merit further biochemical and functional characterization (Figs. 4c–e). Mutating residues on the optides at the modeled interface reduced or eliminated TEAD binding (Figs. 4f, g). TB1G1 and TB2G1 were produced as soluble optides, but only TB1G1 was monodisperse and stable in solution (Fig. 4h and Supplementary Fig. 3). Using surface plasmon resonance, TB1G1 bound TEAD with an equilibrium dissociation constant ($K_D$) of $31 \pm 2$ nM (Fig. 4h,

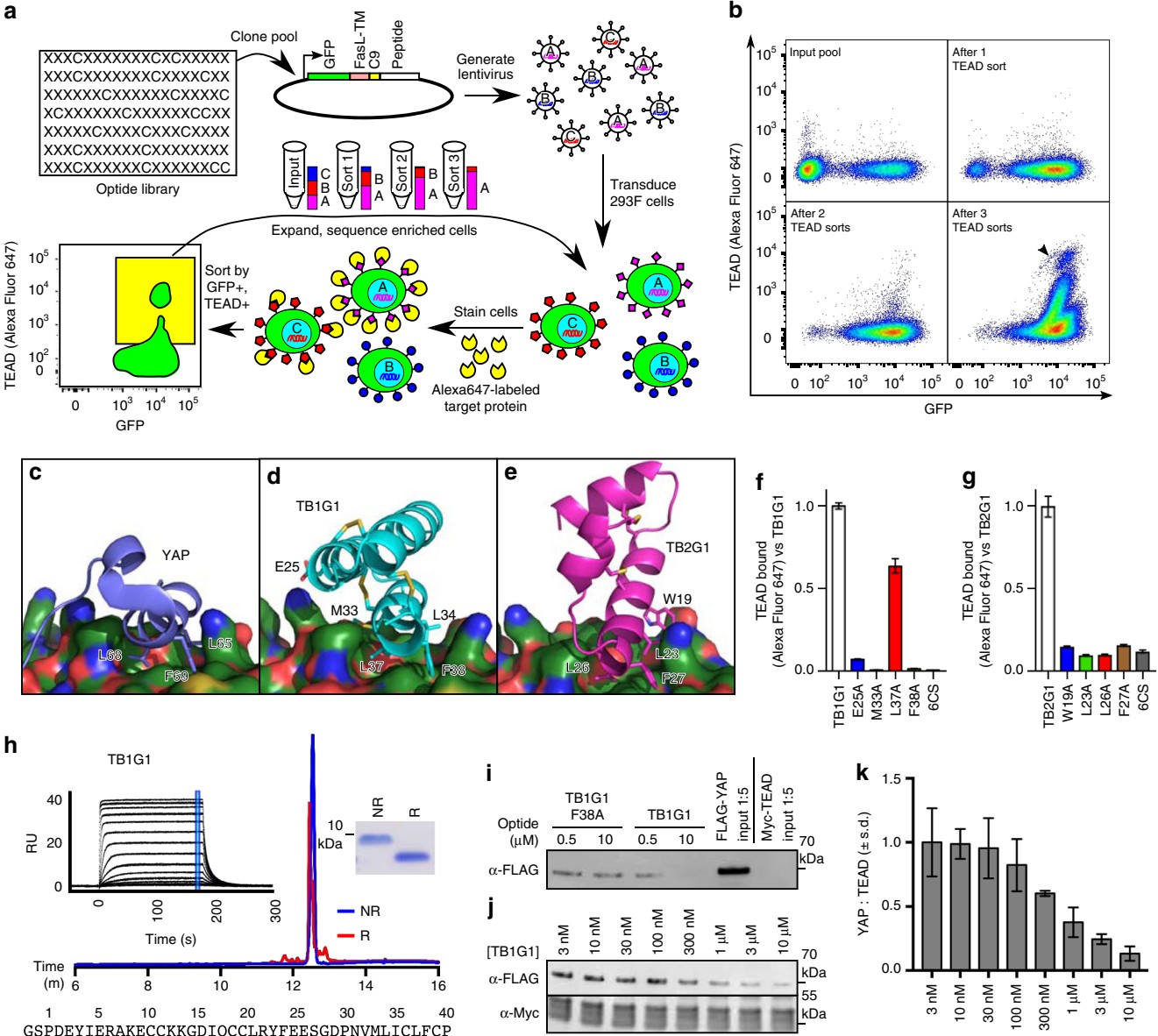

**Fig. 4** Mammalian surface display screening identifies TEAD-binding optides. **a** Scheme for the screening strategy. FasL-TM: transmembrane domain of the FasL protein. **b** Flow profiles (GFP vs TEAD-streptavidin-Alexa Fluor 647) of the library of designed optides in SDGF; shown are the profiles of the unsorted input library (top left), and the library after one (top right), two (bottom left), or three (bottom right) TEAD sorts. Arrowhead in bottom right corresponds to TB1G1 clone within enriched polyclonal pool. **c** YAP:TEAD structure from PDB ID 3KYS, focusing on Interface 2. **d**, **e** TB1G1 **d** and TB2G1 **e** modeled at the same location on TEAD. Both share a conserved "LXXLF" motif with YAP. **f**, **g** The TEAD-binding abilities of 293F cells transfected with SDGF-TB1G1 **f** or SDGF-TB2G1 **g** and interface mutations are shown. 6x Cys to Ser (6CS) to eliminate disulfide bonds were also tested. Binding quantitated by Alexa Fluor 647 levels in transfected cells using flow cytometry. Shown are the normalized median ± 95% confidence intervals from one experiment. **h** Soluble TB1G1 is monodisperse in solution by reversed-phase (RP) HPLC (bottom) and SDS-PAGE (right inset). Reduction with 10 mM DTT was complete and demonstrates a mobility shift in both RP-HPLC and SDS-PAGE. TEAD-binding equilibrium binding constant ($K_D$) was 31 ± 2 nM by SPR (left inset, serial 2-fold dilutions of 2 μM → 0.12 nM in duplicate; RU within blue mark used for curve fit in Supplementary Fig. 8a; see Methods and Supplementary Table 4 for SPR methodology). NR: Non-reduced. R: Reduced with 10 mM DTT. **i** TB1G1 and the poorly-binding F38A variant were tested for inhibition of FLAG-YAP co-immunoprecipition with Myc-TEAD in 293T cell lysate. **j**, **k** Dilution series of TB1G1 in the FLAG-YAP:Myc-TEAD co-immunoprecipitation assay. **j** is one representative blot; **k** is the YAP:TEAD signal ratio, average ± s.d. (N = 4 lanes per dilution from one experiment), quantitated by band densitometry

left inset). Two point mutants at the modeled interface of TB1G1 (L37A and F38A) were also produced, and were indistinguishable from TB1G1 except for increased TEAD-binding $K_D$ (Supplementary Fig. 4 and Supplementary Table 4), with the degree of $K_D$ increase correlating with the reduction in surface TEAD staining (Fig. 4f). Finally, TB1G1 (but not TB1G1-F38A) demonstrated dose-dependent inhibition of YAP:TEAD binding in co-immunoprecipitation experiments (Figs. 4i–k).

**Platform flexibility facilitates rapid affinity maturation.** The concentration used to screen for TEAD binders (200 nM) is similar to that commonly used for yeast display screening[29,45], and under such conditions, cells displaying TB1G1 stain extremely well (~100x background staining; Fig. 4b, arrowhead). However, we wished to investigate the sensitivity of the staining under conditions of increased stringency, by reducing both the concentration and the avidity of the interaction. TB1G1 served as

a good model to test the dynamic range of the surface display platform, varying target concentrations (64 pm to 200 nM) and avidity (tetravalent, bivalent, or monovalent) (Fig. 5a). The TEAD used for staining is both 6xHis-tagged and biotinylated. Hence, avidity was modulated as follows: 1-step co-incubation of TEAD and streptavidin for tetravalent staining; 1-step co-incubation of TEAD and anti-6xHis antibody for bivalent staining; and 2-step incubation, first with TEAD followed by pelleting and resuspending in solution with streptavidin, for monovalent staining. From such variation in staining conditions, TEAD binding was

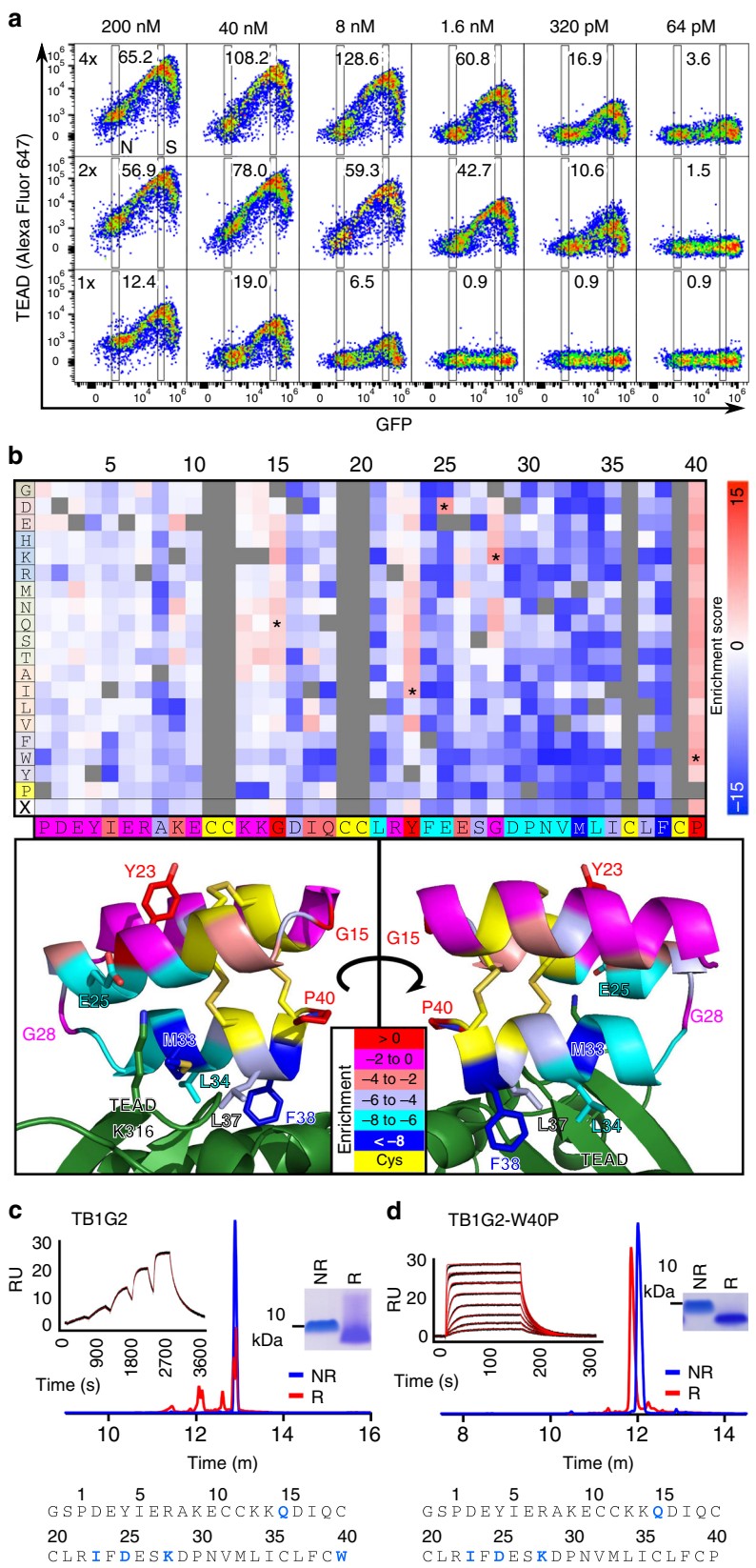

detectible even at 64 pM, with peak signal:noise at 8 nM, under tetravalent staining. Higher valence significantly improved staining; signal:noise was roughly equivalent between 40 nM monovalent and 320 pM tetravalent staining. These specific parameters are likely peptide-specific, but the assay's sensitivity at and below 1.6 nM compares favorably with concentrations used in conventional yeast display screens of CDPs[29,45], possibly owing to the increased surface area of mammalian cells.

For affinity maturation of TB1G1, we used a monovalent, two-step incubation with 20 nM biotinylated TEAD and streptavidin-Alexa Fluor 647. Variation was achieved by site saturation mutagenesis, making a library of every possible non-cysteine substitution. By analyzing the variants' enrichment or depletion after only two rounds of sorting (Supplementary Fig. 5), we identified substitutions conferring improved or reduced TEAD binding (Fig. 5b). Mutation-tolerance of each residue was in agreement with the modeled interaction, suggesting that the design process had engineered a TEAD-binding surface on TB1G1 as intended, and that the methods allow for the detection of both extreme and subtle changes in target binding. We selected five enriched substitutions for further testing (G15Q, Y23I, E25D, G28K, and P40W), which were combined in every possible triple (10), quadruple (5), or quintuple (1) mutant permutation (Supplementary Fig. 6). All but one demonstrated improved surface display TEAD binding. Furthermore, TB1G1 variants that retain the native P40 showed substantial loss of staining when cells were given an additional rinse, suggesting that P40W slows (improves) the optide's off-rate.

The quintuple mutant from the permutation analysis (called TB1G2) and its reversion mutant from W40 back to P40 (TB1G2-W40P) were produced as soluble optides. Both were mono-disperse and stable in solution (Figs. 5c, d) with greatly improved TEAD binding compared to TB1G1 (Supplementary Table 4). The on-rates of both variants were comparable, but the off-rate of TB1G2-W40P was substantially faster (~15-fold) than that of TB1G2, in agreement with the loss of surface binding after extra rinsing (Supplementary Fig. 6c).

**TB1G2 is drug-like and potently inhibits YAP:TEAD binding.** The mammalian display platform is intended to identify drug-like peptides, so we evaluated the stability of TB1G2 under physiological or more extreme conditions. Treatment of TB1G2 with 10 mM DTT produced multiple peaks in RP-HPLC (Fig. 5c), which is unusual for a CDP. Mass spectrometry confirmed incomplete reduction under these conditions (Fig. 6a), while milder, intracellular reducing conditions (10 mM glutathione) had no effect on TB1G2 stability, either soluble (Fig. 6b) or

surface displayed (Fig. 6c). We also tested its protease stability in surface display, where large amounts (40 µg mL$^{-1}$) of trypsin or chymotrypsin produced no change in anti-6xHis staining of 6xHis-tagged TB1G2 (Fig. 6d). Solution thermostability assays, by circular dichroism (Figs. 6e, f) and dye-based thermal shift (Fig. 6g), produced no evidence of altered TB1G2 structure up to 95 °C.

To verify the ability of TB1G2 to disrupt the YAP:TEAD interaction, we performed a TEAD competitive binding assay in the surface display system. This was chosen over co-immunoprecipitation because of improved sensitivity under conditions of low TEAD concentration. 293F cells expressing SDGF-YAP were pre-incubated with varying concentrations of TB1G1 or TB1G2, and then stained with 5 nM TEAD. Both optides inhibited YAP:TEAD-dependent cell staining (Figs. 7a–g), with TB1G2 demonstrating much higher potency.

We next tested for YAP:TEAD inhibition in cells. Bypassing the oxidative secretory pathway, mCherry-T2a-FLAG-TB1G1 and mCherry-T2a-FLAG-TB1G2 were expressed in the cytosol of 293T cells co-transfected with YAP and a TEAD luciferase reporter. T2a-cleaved peptides were not visible by western blot (Supplementary Fig. 7), but reporter activity was reduced ($P = 0.003$ by two-tailed T-test) by mCherry-TB1G2 (Fig. 7h). Furthermore, the fusion proteins show a subtle, cysteine-dependent mobility shift in SDS-PAGE upon reduction (Fig. 7i), suggesting that, when stabilized by mCherry, the peptides have favorable thermodynamic folding to allow cytosolic disulfide formation.

To see whether free TB1G2 could inhibit intracellular YAP:TEAD binding, without direct cytosolic expression or a fusion partner, purified TB1G2 was tested on HeLa cells. We failed to demonstrate cell penetration of TB1G2, so optides were co-administered with dfTAT, a small dimeric peptide that facilitates endosomal escape of cargoes[46] (Figs. 7j, k). To quantitate any change in YAP:TEAD dimerization, we treated HeLa cells with dfTAT and/or optides (5 µM each or PBS) for 90 min, and then performed a proximity ligation assay[47] using primary antibodies against YAP and TEAD. The assay creates visible speckles where YAP and TEAD are in close proximity, and quantitating the speckles per nucleus (Figs. 7l–p) demonstrated a significant (except otherwise noted, $P < 0.0001$ by two-tailed Kolmogorov–Smirnov test) reduction in speckles in cells treated with TB1G2 and dfTAT versus cells treated with dfTAT alone, TB1G2 alone ($P < 0.01$), dfTAT and TB1G1-F38A, or dfTAT and TB1G1. These in vitro and cell-based assays demonstrate the ability of the platform to identify a target-binding CDP with predictable function, and then improve its potency and stability to that of a promising clinical development candidate.

**Fig. 5** Mammalian display saturation mutagenesis to identify substitutions that improve or reduce binding. **a** TB1G1:TEAD binding in surface display tested over a dilution series of biotinylated, 6xHis-tagged TEAD + stain (200 nM → 64 pM), and with different staining methods conferring different avidities. Top row: 1-step co-incubation of TEAD and streptavidin for tetravalent staining (4x). Middle row: 1-step co-incubation of TEAD and anti-His antibody for bivalent staining (2x). Bottom row: 2-step incubation, first with TEAD followed by pelleting and resuspending in solution with streptavidin, for monovalent staining (1x). Signal:noise ratios, quantitated by Alexa Fluor 647 levels in transfected (signal, S) vs. untransfected (noise, N) cells using flow cytometry, are at the top of each plot. **b** Mutational tolerance of TB1G1, evaluated by saturation mutagenesis. Heat map displays enrichment scores for every possible non-Cys substitution in TB1G1, tested for TEAD binding in SDGF surface display screening (20 nM, monovalent staining). Rows are amino acid substitutions, grouped by chemical category. Columns are the TB1G1 protein sequence, duplicated below the heat map. Enrichment score represents a variant's fold-change in population abundance after two rounds of TEAD sorting versus its input abundance, normalized to TB1G1, and $\log_2$-transformed. Warm colors are enriched variants (improved binding); cool colors are depleted. X: Average enrichment score. The TB1G1 sequence and structures are color coded for the average enrichment scores of each residue, with warm colors indicating tolerance to substitution and cool colors indicating intolerance. Asterisks in the heat map indicate mutations combined to create TB1G2. **c, d** Soluble TB1G2 and TB1G2-W40P were analyzed by RP-HPLC (bottom) and SDS-PAGE (right insets) in either non-reducing (PBS) or reducing (10 mM DTT) conditions. TEAD-binding $K_D$ values were 368 ± 4 pM (**c**, TB1G2; 0.044, 0.133, 0.4, 1.2, and 3.6 nM in singleton) and 3.78 ± 0.05 nM (**d**, TB1G2-W40P; serial 2-fold dilutions of 50 nM → 390 pM in duplicate) by SPR (left insets; SPR responses in black, binding model fits in red). Residues mutated from TB1G1 are in blue, bold font in the sequences. Please see the Methods and Supplementary Table 4 for SPR methodology and analytical models

## Discussion

By leveraging mammalian surface display (a technique that has only been reported for antibody affinity maturation to this point[48,49]), optimizing it for CDP expression, CDPs can now be screened with a greater degree of diversity to facilitate identification of de novo binders for difficult to drug targets. Mammalian cells are rarely used for protein screening efforts because they are thought of as more complex, costly, and time-consuming than lower organisms like phage and yeast. The mammalian peptide display platform largely avoids two of these

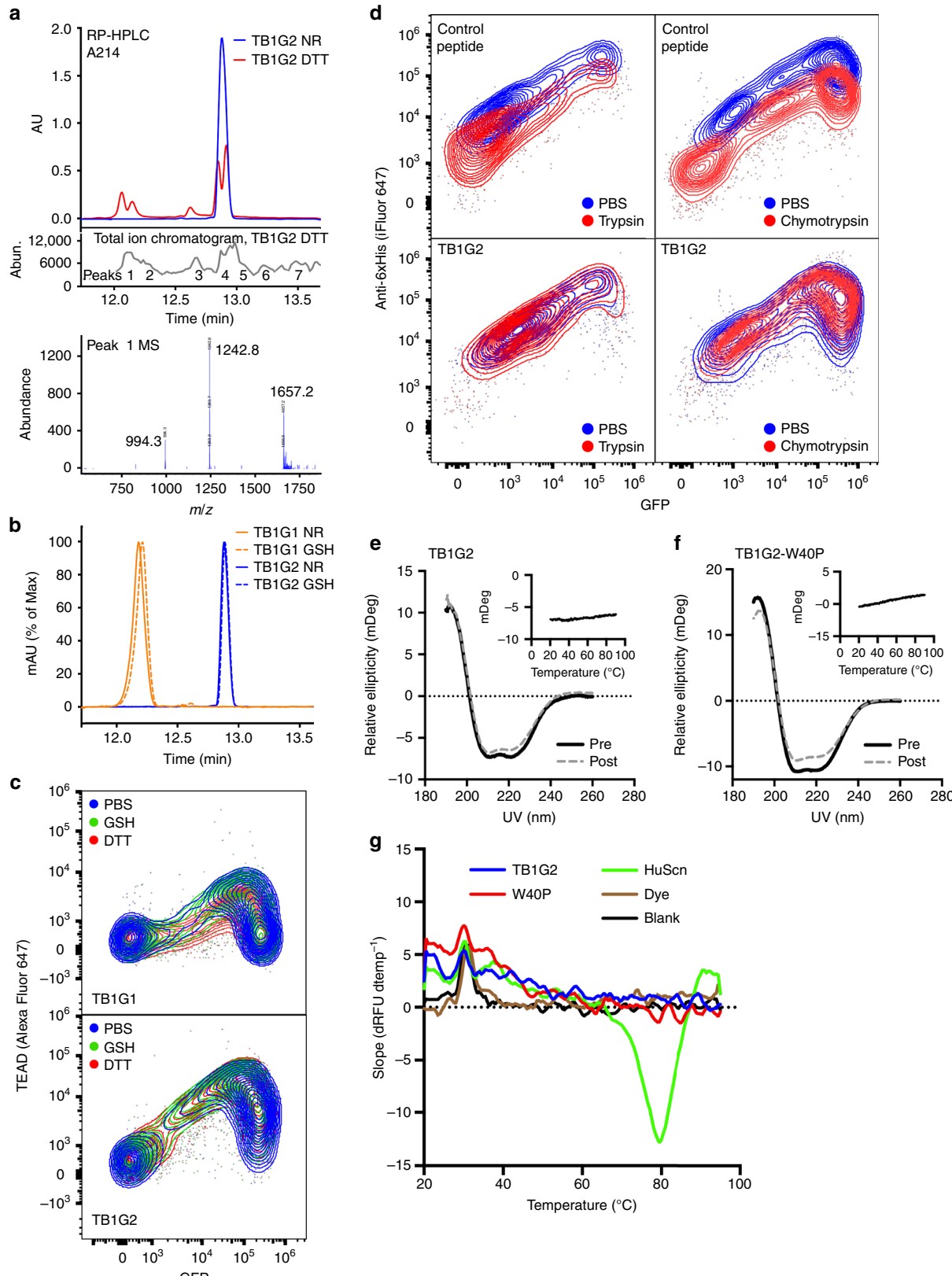

issues. This platform only requires one additional step beyond what is typical for yeast display (direct transformation of yeast vs. viral production and transduction of mammalian cells). Additionally, the differences in costs are mainly limited to tissue culture vs. yeast media, which are dwarfed by the costs of DNA synthesis, sequence analysis, and flow cytometry equipment maintenance.

In this way, the mammalian display platform augments the toolkit available to the CDP screening field, adding to the well-established and successful yeast and bacterial screening methodologies. Drug discovery has innumerable challenges, and incorporating multiple screening paradigms will provide the highest likelihood of finding effective candidate molecules. Here we've shown how the mammalian platform can facilitate the use of diverse libraries containing more challenging CDP scaffolds in routine, cost-effective peptide screening efforts. The flexibility and demonstrable sensitivity of the platform further contribute to its utility. As an additional benefit, one can directly transition from surface display to soluble, endotoxin-free biologics production in the same cell line, allowing for therapeutic candidates to be produced for in vivo testing in the same cell line (and even the same secretory pathway) where their function was first characterized. This reduces the risk that a cross-species or cross-line difference in post-translational modification will reduce a candidate's effectiveness when produced as a soluble product.

Apart from library diversity, one of the challenges for a surface display screening campaign is the generation of a suitable amount of target protein. Many target proteins are difficult to express or solubilize, so being able to screen at dilute concentrations would facilitate investigation of more troublesome, or more expensive, targets. The mammalian platform's sensitivity, with peak signal: noise at 8 nM biotinylated target protein against a first-generation binder like TB1G1, opens up an innumerable range of targets for screening. At this concentration, two 10 μg aliquots (common for commercial recombinant protein vendors) of a 50 kDa biotinylated protein would be sufficient for a full screening campaign, including affinity maturation.

At the same time, for those protein targets available at slightly higher concentrations for screening (40 nM, or 100 μg of a 50 kDa protein for a full campaign), our data have demonstrated that one can select for variants with slower off-rates by altering the staining protocol for reduced avidity. As peptides of this size bind to targets with a relatively small buried surface area, fast-off kinetics are a potential liability of peptide drug candidates. The ability to specifically select for variants with slower off rates can help offset this potential disadvantage.

TB1G2 is an early case study in the benefits and challenges of targeting a cytosolic protein:protein interaction with CDPs. Focusing on peptides without disulfides would eliminate the complication of the cytosolic reducing environment, but the success of disulfide-rich peptides found in nature, including the calcine knottins[24,25], suggests to us that the benefits of disulfide stabilization may outweigh the liabilities. Furthermore, as TB1G2 is resistant to cytosolic reducing conditions, we see that selection for high-affinity target engagement can coincide with selection for reduction resistance. Our favored model for this dual selection is the peptide having extremely low conformational entropy, which would help both affinity (a minimal entropic penalty of binding) and reduction resistance (high local concentration of cysteine sulfhydryls, promoting their interaction[50]). Alternative mechanisms could also play a role.

In TB1G2, we have found a disulfide-stabilized peptide that potently prevents the YAP:TEAD interaction and is resistant to varied insults; however, it is not yet cell penetrant. There are many options for approaching this challenge. First, scaffolds that are naturally cell-penetrant (e.g., the calcines) can be used for binding interface grafting, a technique used effectively on CDP scaffolds to target extracellular proteins[45]. Secondly and better established is attempting to impart cell penetration on an effective peptide, which may be more directly applicable to a candidate like TB1G2. Methods include fusion to known cell penetration motifs (e.g., TAT[51], octa-arginine[52], and penetratin[53]), intra-helical arginine patches[54], or polymeric encapsulation[55], though the formulation must allow the peptide access to the nucleus.

In conclusion, every clade of life makes use of CDPs in drug-like roles. This platform facilitates diversity screening efforts with CDPs, providing a means for identifying candidates to target disease-causing protein:protein interactions that have proven untreatable by conventional means.

## Methods

**High diversity native CDP library selection**. For the identification of diverse native CDPs, we began by filtering protein segments from the January 2014 UniProt[56] database that contained 6, 8, or 10 cysteines within 30–50 amino acids. The resulting CDP motifs were further filtered by the April 2015 ITIS database[57] for taxonomical identification. For laboratory safety compliance, CDP motifs that were annotated as toxins by CDC or FDA guidelines were removed. Finally, we applied taxonomy-weighted random selection (enriching for animal and plant sequences but otherwise attempting to preserve taxonomic diversity) to attain our final library of 9999 members.

**TEAD-binding optide library Rosetta computational design**
*Scaffold construction.* The input topology parameters used for scaffold construction were as follows: minimum and maximum sequence length: 30 and 41 residues, respectively; secondary structure types: helix, helix, helix; secondary structure length ranges: 6–18 residues; turn lengths: 2–4 residues; number of disulfides: 3; disulfide topology: H1-H2, H1-H3, and H2-H3. Several hundred thousand independent design simulations were performed to build a large library of candidate scaffolds, which were then filtered by sequence-structure compatibility, packing, satisfaction of polar groups, and disulfide score. At the start of each design simulation, helix and turn lengths were sampled randomly from the corresponding length ranges, fixing the secondary structure of the design, which was then used to select backbone fragments for a low-resolution fragment assembly simulation. At

**Fig. 6** Second generation TEAD binder has favorable stability. **a** Reversed-phase (RP) HPLC trace of TB1G2 under non-reducing or strongly reducing (10 mM DTT) conditions (top). The sample under reducing conditions was analyzed in an in-line LC/MS mass spectrometer, identifying peaks of interest (middle). Peptide *m/z* of representative peak P1 (bottom) shown, corresponding to a mass of 4971.4 Da. The non-reduced peptide's mass was measured at 4968.7 Da on the same instrument. Full mass spectra available in Supplementary Fig. 10. **b** RP-HPLC of TB1G1 and TB1G2 under either non-reducing (NR) or intracellular reducing conditions using 10 mM glutathione (GSH). **c** 293F cells expressing SDGF-TB1G1 (top) or SDGF-TB1G2 (bottom) were incubated with either PBS, 10 mM glutathione (GSH) or 10 mM DTT for 5 mins before being washed and tested for TEAD binding (20 nM, 2-step stain with Alexa Fluor 647-streptavidin). **d** A control peptide, with known sensitivity to proteases, was cloned into SDPR and incubated with PBS or either 40 μg mL⁻¹ trypsin (top left) or 40 μg mL⁻¹ chymotrypsin (top right), followed by treatment with reducing agent (5 mM DTT) and iFluor 647 anti-6xHis staining. Bottom: Same as top, with cells expressing SDPR-TB1G2. **e, f** Circular dichroism spectra of soluble CDPs TB1G2 **e** and TB1G2-W40P **f** indicate a structure dominated by α-helical elements, and that this secondary structure signature is identical before (Pre) and after (Post) incubation at 95 °C. Insets: relative ellipticity at 220 nm during heating from 20 °C to 95 °C. **g** SYPRO Orange thermal shift assay of optides. Shown is the slope of the change in relative fluorescence units (dRFU dtemp⁻¹) during heating from 20 °C to 95 °C. Human siderocalin (HuScn) produced an expected melting temperature of 79 °C, as interpreted by the peak of its RFU vs temperature slope. Conversely, no melting temperature could be determined for the two optides tested (TB1G2 and TB1G2-W40P)

the end of the low-resolution simulation, the protein backbone was scanned for residue pairs that could be linked by disulfide connections using a library of N-C$_\alpha$-C backbone transforms derived from disulfide bonds in the protein structure database. Backbones with matching residue pairs that satisfied the disulfide topology contraints were used to initiate an all-atom sequence design simulation consisting of two cycles of alternating fixed-backbone sequence design and fixed-sequence structure relaxation. Final designs were filtered for packing (sasapack score < 0.5), satisfaction of buried polar groups (using a 1.0 A probe radius), and sorted by energy per residue. The top 10% of the filtered designs were assessed for sequence-structure compatibility by an in silico refolding test in which the design

sequence is used to initiate 3000 independent structure prediction simulations. Success was measured by assessing the fraction of low-energy structure prediction models within 2Å C$_\alpha$-RMSD of the design model.

*Interface design.* The crystal structure of the YAP:TEAD complex (PDB ID 3KYS) was examined to identify binding patches on TEAD and corresponding backbone elements on YAP to serve as templates for interface design. The following backbone residue segments were selected as superposition targets for orienting design scaf-folds: 3KYS/B/53-55 (Interface 1), 3KYS/B/55-57 (Interface 1), 3KYS/B/64-68 (Interface 2), 3KYS/B/64-69 (Interface 2), 3KYS/B/86-89 (Interface 3), 3KYS/B/94-

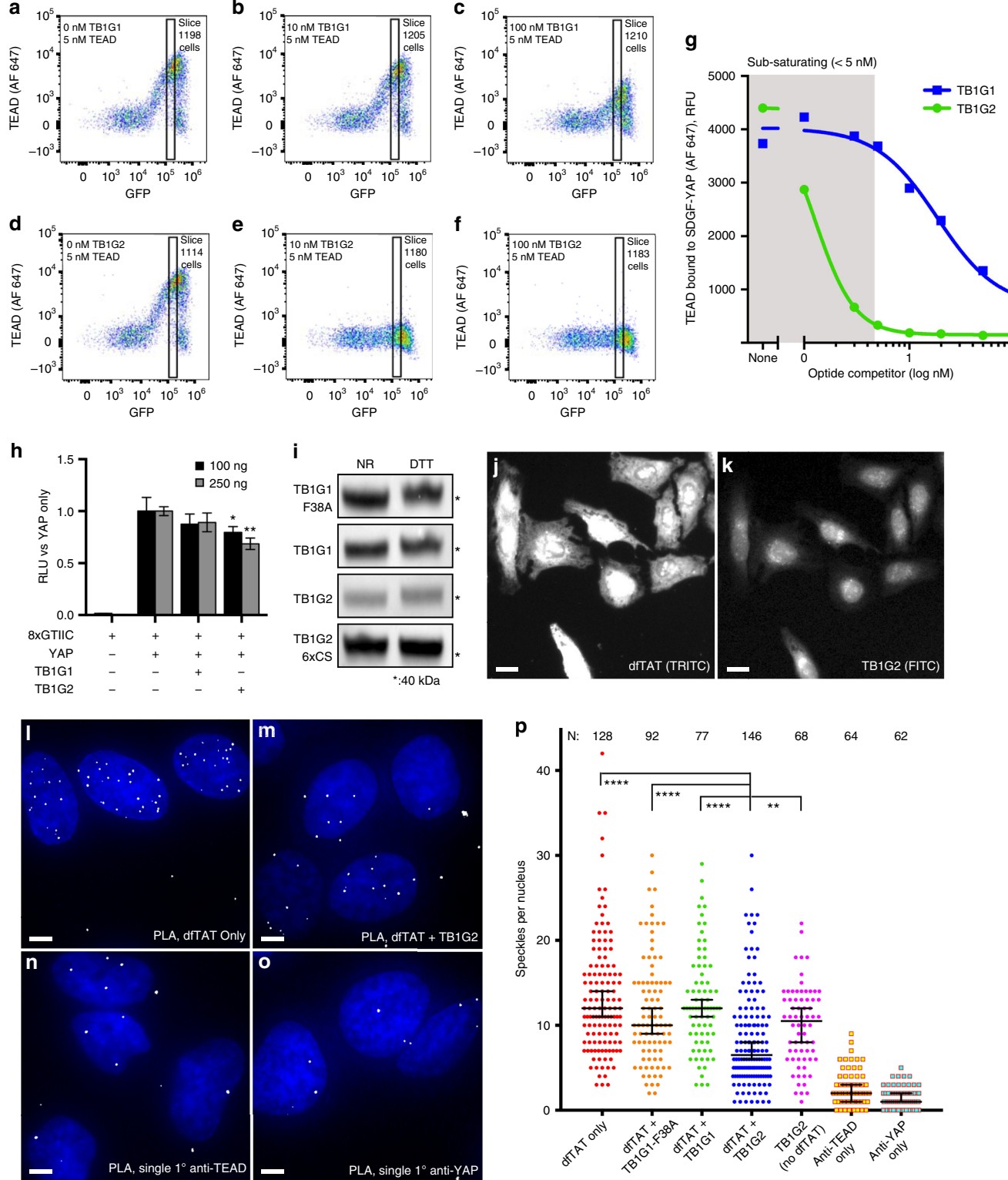

96 (Interface 3) (given as: PDB ID/chain/residue numbers). For each peptide scaffold, 150 design simulations were conducted targeting each YAP backbone segment selected for superposition. Each design simulation consisted of the following steps: (1) superimposing the scaffold backbone onto the YAP backbone segment using a scaffold backbone element with matching secondary structure, (2) random small perturbations to generate diversity and relieve backbone clashes, (3) all-atom sequence design alternating between fixed-backbone sequence selection (amino acid sequence optimization) and fixed-sequence structure relaxation. Final interface designs, in the form of modeled interactions of the CDP variants and TEAD at the respective superposition target sites, were filtered for satisfaction of polar groups (using a 1.0A probe radius), interface surface complementarity (sc score > 0.5), and interface quality (predicted binding energy per 100Å of buried SASA<−1.1), and sorted by predicted binding energy. Top-scoring designs were assessed by an in silico redocking test in which the redesigned scaffold peptide was removed from TEAD, randomly reoriented, and redocked onto the TEAD protein structure. Success was measured as the fraction of low-energy redocking simulations that reached a final state close to the designed interface conformation. In all, the scaffold construction and interface design scripts generated 7533 predicted TEAD binders for the library.

**Biotinylated His-Avi-TEAD production.** Biotinylated, His-Avi-TEAD1(194–411) was produced in Hi5 insect cells using the BacMagic system (EMD Millipore), as per manufacturer protocols. Briefly, the transgene was cloned into the pIEX-BAC3 vector and then cotransfected with BacMagic-3 DNA (100 μg vector, 1 μL Bac-Magic-3) using calfectin II into Sf9 cells in a 12-well plate. Baculovirus encoding His-Avi-TEAD was amplified in Sf9 cells, and viral supernatant (5 mL) used to transduce 2.5E8 Hi5 cells in 250 mL Express Five media supplemented with L-glutamine. Transduced cells were grown for 72 h (expanding to 500 mL) at 27 °C and 140 RPM.

To harvest, cells were pelleted (2000 × g, 10 min) and then resuspended in I-PER buffer (Invitrogen) containing protease inhibitor cocktail (ThermoFisher), benzonase at 1:10 000 (Millipore), 0.5 mM TCEP, and 20 mM imidazole. Lysate was clarified (10 000×g, 30 min), and nickel NTA resin was used to purify GST-TEAD. Protein was buffer exchanged (Zeba spin columns, 5 mL capacity) into thrombin cleavage buffer (25 mM Tris pH 8.4, 150 mM NaCl, 0.1% Triton X100, 10% glycerol, 2.5 mM CaCl₂). Half of the 4 mL eluate was treated with 5 μL restriction grade thrombin (EMD) overnight at room temperature. The TEAD was re-purified on nickel resin and then by FPLC on a Superdex 200 10/300 GL SEC column (GE Healthcare). SEC running buffer contained 10 mM phosphate buffer pH 7.2, 50 mM NaCl, 0.5 mM TCEP. Purified TEAD was biotinylated using the BirA-500 kit (Avidity) as per manufacturer's protocol, followed by a final buffer exchange into PBS containing 5% glycerol, and storage in small aliquots at −80 °C.

**CDP production and purification.** Test CDPs and TEAD-binding optides were cloned into our secreted, soluble protein production vector, DNA-sequence validated (Sanger sequencing, Genewiz; coding DNA sequences are shown below, and raw sequence files supplied as Supplementary Data 1), and purified, as per standard protocols[34,58] at either large (2 L in 5 L flasks) or small (1 mL in 96-well deep well blocks) volumes. In brief: peptide coding sequences were cloned into the Daedalus vector, a lentivector driving secretion of siderocalin-tagged proteins. The siderocalin is 6xHis-tagged and the linker contains a TEV cleavage site, leaving only "GS" behind on the peptide's N-terminus after cleavage. (Note: for peptide amino acid numbering, we begin after this GS, as it is irrelevant to surface display and would otherwise confound notation comparing surface and soluble forms.) VSV-G pseudotyped lentivirus was produced through standard methods, and suspension 293F cells were transduced, after which they were grown in FreeStyle (Thermo-Fisher) expression media. For small scale, $1 \times 10^6$ cells were transduced in 1 mL with 100 μL viral supernatant, shaken at 1000 rpm, with 3 mM valproic acid added after 5 days, until harvest (~7 days). For large scale, $1 \times 10^7$ cells were transduced in

10 mL with 1 mL viral supernatant (target multiplicity of infection is ~10), after which the culture (shaken at 125 RPM) was expanded over the course of 10-12 days to 2L final volume. Peptides were collected from culture media after pelleting cells and 0.22 μm filtration of debris, followed by nickel resin purification and TEV protease cleavage. For large scale preps, additional SEC purification is performed.

Quality control was performed by SDS-PAGE followed by Coomassie staining (large volume only; see Supplementary Fig. 9 for full Coomassie stained gels for the TEAD-binding optides produced for this study), and by reversed-phase HPLC on an Agilent model 1260 with an in-line Agilent 6120 LC/MS. (see Supplementary Fig. 10 for mass spectra of TB1G2). SEC-purified large scale peptides were analyzed with a C18 column for large scale preps, while small scale crude TEV cleavage product was analyzed on an AdvanceBio RP-mAb Diphenyl, 4.6 × 100 mm, 3.5 μm, LC column. TEAD binding was assessed using surface plasmon resonance (see below). All optides were TEV-cleaved and analyzed as independent, soluble proteins. Protein concentrations were determined by UV spectral absorption and/or amino acid analysis. All HPLC peptide analysis was conducted at a wavelength of 214 nm.

Full Sanger sequencing results for Daedalus-expressed optides are in Supplementary Table 5, with the raw trace files available as Supplementary Data 1. Coding DNA sequence of the soluble TEAD binding optides were as follows (between, and including, the relevant cloning *Bam*HI and *Not*I cut sites of the Daedalus vector):

TB1G1:
5′-GGATCCCCTGATGAATATATTGAACGCGCCAAAGAATGCTG
CAAAAAAGGCGATATTCAGTGCTGCCTGCGCTATTTCGAAGAATCC
GGGGACCCCAA CGTGATGCTGATTTGCCTGTTCTGCCCCTAATGC
GGCCGC-3′

TB1G1-L37A:
5′-GGATCCCCAGATGAATATATTGAGCGCGCAAAAGAATGCTGC
AAAAAAGGTGATATTCAGTGCTGCCTGCGCTATTTCGAGGAATCT
GGGGACCCTAACGTGATGCTGATTTGCCGCCTTCTGCCCCTAATGCG
GCCGC-3′

TB1G1-F38A:
5′-GGATCCCCTGATGAATATATTGAACGCGCCAAAGAATGCTGCAAA
AAAGGCGATATTCAGTGCTGCCTGCGCTATTTCGAAGAAAGTGGTGA
CCCCAACGTGATGCTGATTTGCCTGGCCTGCCCCTAATGCGGCCGC-3′

TB1G2:
5′-GGATCCCCCGATGAATATATTGAACGCGCCAAAGAATGCTGCAAA
AAACAGGATATTCAGTGCTGCCTGCGCATTTTCGATGAAAGCAAAGAT
CCCAACGTGATGCTGATTTGCCTGTTCTGCTGGTAATGCGGCCGC-3′

TB1G2-W40P:
5′-GGATCCCCCGATGAATATATTGAACGCGCCAAAGAATGCTGCAA
AAAACAGGATATTCAGTGCTGCCTGCGCATTTTCGATGAAAGC
AAAGATCCCAACGTGATGCTGATTTGCCTGTTCTGCCCCTAATGCG
GCCGC-3′

**SDGF surface display vector construction and cloning.** Our primary surface display vector, SDGF, was based on the Daedalus vector[34], with three changes. (1) A simian CMV promoter[59] was used instead of the standard CMV promoter. (2) Siderocalin was replaced with a fusion protein containing (N-term to C-term): GFP; a GGGS spacer; the transmembrane domain (residues 72–110) of the single pass Type II transmembrane protein FasL (human Fas ligand); a 2x-GGGS spacer; a 9-residue bovine rhodopsin antigen (TETSQVAPA); a 2x-GGGS spacer; a TEV-cleavage site (ENLYFQGGS), which includes a GS-linker; and the displayed protein. (3) The IRES-GFP is removed. The entire construct is cloned between conventional lentiviral SIN LTRs, and introduced to cells by transient transfection or lentiviral transduction. A variant with mCherry substituting GFP also exists, called SDRF. Studies using protease digestion use a modified vector (SDPR) that only differs from SDGF in the mutation of Lys, Arg, Phe, Trp, and Tyr residues

**Fig. 7** TB1G2 potently prevents TEAD from binding to surface-displayed and intracellular YAP. **a–f** 293F cells expressing SDGF-YAP were exposed to 5 nM biotinylated TEAD and 0-100 nM soluble TB1G1 **a–c** or TB1G2 **d–f** before being washed and incubated with 5 nM streptavidin-Alexa Fluor 647. Binding was then assessed by flow cytometry, and quantified using the Alexa Fluor (AF) 647 values of cells within the narrow "slice" gate. **g** Median TEAD (5 nM) binding to cells expressing SDGF-YAP in the presence of TB1G1 or TB1G2. Note: optide concentrations below 5 nM (shaded region) are below the TEAD concentration and are therefore non-saturating. **h** TB1G1 and TB1G2 were expressed in 293T cytosol as part of an mCherry-T2a-optide fusion (100 or 250 ng plasmid), co-transfected with YAP and 8xGTIIC (TEAD luciferase reporter) plasmids. RLU: relative luminescence units. *: $P < 0.05$, **$P < 0.005$ vs. YAP only, $N = 3$ wells, 4 measurements per well; $P$-values from two-tailed $T$-test. Data presented as average ± s.d. from one experiment. **i** Lysates with FLAG-tagged optides in the mCherry-T2a-optide construct were tested for SDS-PAGE mobility shift upon DTT reduction using anti-FLAG western blot. Shown are the uncleaved fusion protein bands. 6xCS: all six cysteines were mutated to serines. **j, k** 5 μM TB1G2-DyLight 488 was introduced into HeLa cells using 5 μM dfTAT. **l–o** Proximity ligation assay (PLA) in HeLa cells, using anti-YAP and anti-TEAD, produces speckles overlapping DAPI-stained nuclei. Shown are representative images of cells treated with 5 μM dfTAT alone (**l**) or 5 μM dfTAT and 5 μM TB1G2 (**m**). Control PLA reactions that omit either anti-YAP **n** or anti-TEAD **o** show non-specific speckles. **p** Automated counting of YAP:TEAD PLA speckles per nucleus was performed on HeLa cells treated with 5 μM dfTAT and/or 5 μM TEAD-binding optides. Each dot represents a single nucleus, with the bars representing the median ± 95% confidence intervals. **$P < 0.01$. ****$P < 0.0001$. $P$-values determined by two-tailed Kolmogorov–Smirnov test. Automated counts combined from two complete experimental replicates. Scale bars: 20 μm (**c, d**) and 5 μm (**e–h**)

from the C-terminal portion of the FasL transmembrane domain and from the TEV-cleavage site (which removes trypsin and chymotrypsin sensitive sites), as well as the addition of a C-terminal GGGS-6xHis (HHHHHH) tag. Protein/peptide coding sequences are inserted between unique BamHI and NotI cut sites for SDGF, and between unique *Bam*HI and *Age*I cut sites for SDPR. DNA was sourced from multiple vendors: IDT for single constructs; CustomArray, Inc. and Twist Bioscience for pooled oligonucleotide libraries. In either case, constructs were ordered to include flanking PCR primer sites, so cloning was performed by PCR amplifying coding sequences with the appropriate homologous overhangs allowing for assembly by any number of strategies (restriction digestion, In-Fusion, or Gibson Assembly) based on reagent availability. DNA was PCR-amplified (SDGF Forward primer: 5′-TGTACTTCCAGGGAGGATCC-3′; SDGF Reverse primer: 5′-AATGGTGATGAGCGGCC-3′; SDPR Forward primer: 5′-CCAGCAGGAGGTG-GAAGCG-3′; SDPR Reverse primer: 5′-ATGATGGTGATGATGGTGA-GATCCTC-3′) and, in the case of oligonucleotide pools, gel purified prior to cloning. Care was taken to ensure DNA was not over-amplified, resulting in aberrant self-priming due to primer depletion. All cloned protein sequences were confirmed by Sanger sequencing (Genewiz). Cloning was accomplished either by restriction digestion (*Bam*HI and either *Not*I or *Age*I, NEB; T4 DNA Ligase, Invitrogen), In-Fusion (Clontech), or Gibson Assembly (NEB), with all transformations using Stellar chemically competent cells (Clontech).

**Mammalian surface display.** Expression analysis of the diverse CDP library began with cloning the pooled oligonucleotide library into SDPR, followed by lentivector production (VSV-G pseudotyped, produced by standard methods in 293T cells using the envelope plasmid pMD2.G and the packaging plasmid psPAX). Ten million cells were transduced with the vector at a multiplicity of infection (MOI) of 1, followed by a 3-day incubation (125 RPM shaking incubator, otherwise standard tissue culture conditions). Cells were then pelleted (500x*g*, 5 min) in two separate 10 million cell aliquots, and resuspended in 1.5 mL PBS with either 0 or 5 μg mL$^{-1}$ sequencing-grade trypsin, and incubated at RT for 5 min. After incubation, 6 mL PBS containing 12.5 mM DTT were added, and cells were incubated on ice for 5 min. 7.5 mL Flow Buffer (PBS with 0.5% BSA and 2 mM EDTA) were added, and the cells were pelleted (500x*g*, 5 min). Pellets were resuspended with 3 mL Flow Buffer containing 8 mM iFluor 647 anti-6xHis antibody (Genscript Cat. # A01802-100) and 1 μg mL$^{-1}$ DAPI, and incubated on ice for 30 min with gentle agitation. After incubation, cells were diluted with 9 mL Flow Buffer, pelleted (500x*g*, 5 min), and resuspended in 2 mL Flow Buffer prior to sorting (BD Aria 2 flow sorter). Additional information concerning sorting parameters and Illumina DNA sequence analysis to render surface protein content quantitation is available in the Supplementary Methods.

Singleton candidate testing took place in transfected suspension HEK-293 cells. Briefly, cells were transfected by adding 2.5 μg SDGF vector and 3.5 μg polyethyleneimine to $2 \times 10^6$ cells in 1 mL media, in a 24-well suspension tissue culture dish. Cells were incubated for 2–3 days at 37 °C at 140 RPM and 8% CO$_2$ in FreeStyle media (ThermoFisher), splitting 1:1 daily. All pooled screening took place in 293F cells transduced with lentivirus delivering the SDGF construct at an MOI of 1.

Unless otherwise indicated in the text or figure legends, staining for target binding took place as follows. Transduced or transfected suspension HEK-293 cells were pelleted (500x*g*, 5 min) and resuspended at up to $8 \times 10^6$ cells per mL in Flow Buffer containing DAPI and target protein, with or without Alexa Fluor 647-conjugated streptavidin (ThermoFisher) in equimolar quantity to the target protein. All staining incubations took place at 4 °C for 30 min with mild agitation. For the initial assay validation and pooled screening, 200 nM biotinylated TEAD was used, pre-mixed with streptavidin. Cells were incubated, diluted 4-fold with Flow Buffer, pelleted (500x*g*, 5 min), and resuspended in fresh Flow Buffer (up to $6.5 \times 10^6$ cells per mL) before flow cytometry. For the SSM maturation, only 20 nM biotinylated TEAD was used, and cells were incubated with TEAD alone, diluted 4-fold with Flow Buffer, pelleted, and then resuspended with 20 nM streptavidin. Cells were incubated again, diluted 4-fold with Flow Buffer, pelleted (500x*g*, 5 min), and resuspended in fresh Flow Buffer (up to $6.5 \times 10^6$ cells per mL) before flow cytometry. The "Extra Wash" protocol is identical to the SSM maturation protocol, except that an additional Flow Buffer-only wash step is included immediately after the 30 minute target protein staining. Flow sorting took place on a Beckton-Dickinson Aria II (See Supplementary Fig. 11 for an example of the flow sorting gating process), while analysis took place on a combination of Beckton-Dickson LSR II and Acea NovoCyte flow cytometers. Data analysis was performed on FlowJo (FlowJo, LLC), Excel (Microsoft), Prism 7 (Graphpad), and Matlab R2015a (MathWorks).SDPR protease resistance testing used sequencing grade enzymes from Promega: Trypsin, Cat. # V5111; Chymotrypsin, Cat. # V1062.

**Next generation sequencing.** For surface expression analysis and binding screens, enrichment or depletion of variants were assessed by Illumina sequencing. Briefly, cell pellets ($1.5 \times 10^6$ cells, 3 technical replicates) were resuspended in 50 μL Terra Direct PCR Mix (Clontech) and amplified for 16 cycles using the original cloning primers. Up to four aliquots were then diluted 16-fold into 60 μL Phusion DNA Polymerase reactions and amplified using distinct Illumina primers, containing adaptor sequences for flow cell adherence. Forward primers also included a 6 bp in-line barcode for multiplexing up to 15 samples per lane. Samples were run on an

Illumina HiSeq 2500 in rapid mode. Bowtie2 software was used for mapping. Scoring was default, with the following two exceptions: frameshifts were assigned an artificially high penalty (500), because any frameshift could destroy peptide structure with minimal effect on DNA mapping; and for saturation mutagenesis, a perfect match was required to map due to the close sequence homology of all members in the library. Excel 2011 (Microsoft) and MATLAB R2015a (Math-Works) were used for data processing and analysis. The variant enrichment score heat map shown in Fig. 5b is available in tabular format as Supplementary Data 2.

**High diversity native CDP library data processing.** Please see the Supplementary Methods for information on how Next Generation Sequencing data of the surface content screen was processed for quantitation. Supplementary Data 3 contains the following information for each of the 9999 library members: raw reads per sample (the nature of all samples being described in the Supplementary Methods), protein content score, trypsin resistance score, peptide fraction of whole native protein, HPLC classification, presence or absence of glycosites, annotation as knottin/defensin or not, phylogenetic kingdom and class, and QC threshold values. The methods and data documents should allow reproduction of the data used in relevant figures, but the sequences, in the form of DNA or protein for each CDP, were withheld as proprietary information. For the purposes of quality control, read abundance thresholds were chosen semi-arbitrarily (described in the Supplementary Methods), limiting analyzed data to those 4298 peptides for which the sequence data allows confident quantitation, as validated by high concordance between the independent runs ($R^2 > 0.6$ for both untreated and trypsin-treated protein content scores). The full unthresholded data is included in the aforementioned supplement.

**Surface plasmon resonance interaction analyses.** SPR experiments were performed at 25 °C on a Biacore T100 instrument (GE Healthcare) with Series S SA chips using a running buffer of HBS-EP + (10 mM HEPES, pH 7.4, 150 mM NaCl, 3 mM EDTA, 0.05% surfactant P20) with 0.1 mg mL$^{-1}$ bovine serum albumin. Biotinylated TEAD at 2 μg mL$^{-1}$ was injected over a flow cell at 10 μL min$^{-1}$ to capture ~300 SPR response units (RUs). A reference surface was generated by capturing a molar equivalent of biotinylated human transferrin receptor ectodomain. For analytes which could reach steady-state, serial 2-fold dilutions were prepared in running buffer at concentration ranges which widely spanned each optide's $K_D$. Duplicate samples, interspersed with multiple buffer blanks, were randomly injected at 50 μL min$^{-1}$ with 2–5 min of association and 2–5 min of dissociation. Regeneration was accomplished with buffer flow alone. Double-referenced data were fit with either a 1:1 steady-state or kinetics binding model using BIAevaluation 2.0.4 software (GE Healthcare). TB1G2 was run using a single cycle kinetics protocol in the T100 Control 2.0.4 software, as this sample did not reach steady-state and did not dissociate completely with buffer flow alone over a reasonable amount of time for classical kinetic analysis. Serial 3-fold dilutions (3.6 nM to 0.044 nM) of this optide were prepared in running buffer and injected at 50 μL min$^{-1}$ in increasing concentration order with 7 min of injection time and 15 min dissociation. Two buffer blank cycles for referencing were run prior to analyte injection and one buffer blank cycle followed which allowed time for complete analyte dissociation prior to the next analyte injection. Double-referenced data were fit with the 1:1 binding model for single cycle kinetics using BIAevaluation 2.0.4 software (GE Healthcare). Figures were made in Prism 7 (GraphPad) for Mac OS X version 7.0a. SPR measurements are presented with error in the text and figures, but note that this error represents a precision estimate based on fitting residuals, rather than an accuracy estimate based on replicate measurements. Please see Supplementary Table 4 for more specific methodology used for each optide.

**YAP:TEAD disruption and co-immunoprecipitation.** 293T cells were transfected (TransIT-LT1, Mirus) with either the Myc-tagged TEAD expression plasmid pRK5-Myc-TEAD1 (Addgene plasmid # 33109) or the FLAG-tagged YAP expression plasmid pFLAG-YAP1 (Addgene plasmid # 66853). After 2 days of growth, cells were lysed with RIPA buffer (ThermoFisher). TEAD lysate was bound to anti-Myc agarose resin (Sigma), which was then incubated for 30 min at 4 °C with 50 μL FLAG-YAP-transfected cell lysate (pre-mixed with competitive TEAD-binding Optides), in a final volume of 100 μL. Resin was then washed twice with 500 μL PBS, and then resuspended in 20 μL 2x LDS sample buffer. Beads were then boiled before SDS-PAGE and western Blot (anti-FLAG M2, Sigma F3165, at 1:2000; anti-Myc tag, Abcam ab9106, at 1:2000; LiCor donkey anti-mouse 925–32212 and goat anti-rabbit 925-68071 at 1:10 000). See Supplementary Fig. 12 for full blots corresponding to Figs. 4i–k.

**Circular dichroism.** Protein secondary structures were assessed using circular dichrosim (CD). CD spectra were measured with a Jasco J-720W spectro-polarimeter using a 1.0 mm path length cell. Protein samples (25-30 μM) in 10 mM phosphate buffer (pH 7.4) were analyzed at a wavelength range of 190 to 260 nm. To determine protein thermal stability, samples were subjected to an incremental increase in temperature (2 °C per min, 20 to 95 °C), and stability and protein unfolding were monitored at 220 and 215 nm for α-helix and β-sheet secondary structures, respectively. Data are expressed in terms of relative ellipticity [θ], reported in mdeg.

**Thermal shift assay.** Protein melting temperature ($T_m$) determination was performed by monitoring protein unfolding using SYPRO Orange dye (Molecular Probes) as described[60]. In brief, 0.1 mg mL$^{-1}$ protein samples in 20 μL total volume PBS buffer were mixed with 2 μL of 10× SYPRO Orange dye. Dye intercalation into the hydrophobic protein core following protein unfolding was assayed using the C1000 Touch Thermal Cycler with CFX96 Deep Well Real-Time System (BioRad). Samples were heated from 20 to 95 °C with stepwise increments of 0.5 °C per min and a 5 s hold step for every point, followed by fluorescence reading. $T_m$ were calculated by analyzing the derivatives of Relative Florescence Units (RFU).

**Cytosolic optide expression.** FLAG-tagged optides were cloned into a mammalian expression vector consisting of a CMV promoter, then a monocistronic mCherry-T2a-peptide sequence. These were transfected, along with (where indicated) 50 ng pFLAG-YAP1 and 300 ng 8xGTIIC-luciferase (Addgene plasmid #34615) into 3 × 24-well plate wells of 293T cells. 24 h post-transfection, cells were either harvested for luminescence (ONE-Glo, Promega) or western blot (anti-FLAG M2, Sigma).

**Protein transfection with dfTAT reagent.** dfTAT was a kind gift of Jean-Philippe Pellois (Texas A&M). Solution was diluted to 50 μM in PBS for a 10× working stock, for a final concentration of 5 μM in culture wells. TB1G2, TB1G1, and TB1G1-F38A were fluoresceinated with DyLight 488 NHS-ester (ThermoFisher), with final dye:Optide labeling ratios between 0.8 and 1.4 as assessed by A280 and A488 on a Nanodrop spectrometer (Thermo Scientific). For confirmation of Optide cell penetration, HeLa cells were plated in a 96-well plate in DMEM with 10% FBS, 1× penicillin / streptomycin (Pen/Strep) and grown overnight to ~50% confluence. Cells were gently washed three times with PBS containing 1 mM CaCl$_2$ and 0.5 mM MgCl$_2$, then two times in serum-free DMEM. Wells (50 μL total) received either 5 μM (final) Optide in PBS, or PBS alone, and also received either 5 μM (final) dfTAT reagent in PBS, or PBS alone, before a 60 min incubation at 37 °C and 5% CO$_2$ in a humidified tissue culture incubator. After incubation, cells were gently washed three times in PBS, followed by fixation at 4 °C for 10 min with 4% formaldehyde in PBS. Fixed samples were washed three times with cold PBS, then permeabilized at room temperature for 10 min with 0.25% Triton X-100 in PBS; permeabilization was done for consistency with later proximity ligation assays. Samples were rinsed three more times in PBS prior to imaging on an Evos FL microscope (Life Technologies) with a ×20 objective. Images were processed in ImageJ for brightness/contrast adjustment.

**Proximity ligation assay.** TEAD-binding Optides used (TB1G1-F38A, TB1G1, and TB1G2) were a 1:1 mix of unaltered Optide and Optide reacted with DyLight 488 NHS-ester, as above. HeLa cells were seeded in an eight-well chamber slide (Nunc Lab-Tek II) in DMEM + 10% FBS and 1× Pen/Strep, and grown overnight, reaching ~50% confluence. Cells were gently washed three times with PBS containing 1 mM CaCl$_2$ and 0.5 mM MgCl$_2$, then two times in serum-free DMEM. Wells (120 μL total) received either 5 μM (final) Optide in PBS, or PBS alone, and also received either 5 μM (final) dfTAT reagent in PBS, or PBS alone, before a 90 min incubation at 37 °C and 5% CO$_2$ in a humidified tissue culture incubator. After incubation, cells were gently washed 3× in PBS, followed by fixation at 4 °C for 10 min with 4% formaldehyde in PBS. Fixed samples were washed three times with cold PBS, then permeabilized at room temperature for 10 min with 0.25% Triton X-100 in PBS. Samples were rinsed three more times in PBS prior to proximity ligation assay (PLA).

PLA was done using the Duolink In Situ Fluorescence (Far Red) kit according to manufacturer's instructions, using supplied buffers without substitution and at recommended volumes for 1 cm$^2$ samples. All incubations took place at 37 °C in a humidified chamber. Samples were blocked for 30 min, followed by incubation for 1 h with primary antibodies against human YAP (1:100 rabbit anti-YAP1, AbCam catalog number ab52771) and/or human TEAD (1:200 mouse anti-TEF-1, BD Biosciences catalog number 610923). Samples were washed twice (well dividers were removed after these washes, and further washes took place in a Coplin jar), and then ligation performed for 30 min. After ligation, samples were washed twice prior to the 100 min amplification reaction. Slide then was washed, briefly dried, and mounted with supplied mounting media and a coverslip. Imaging took place on a DeltaVision Elite (GE) with a 40x objective, and complete Z-stacks were acquired and deconvolved. For quantitation of nuclear speckles in ImageJ, UV and Cy5.5 filtered images were Z-projected (maximum intensity), and processed using a custom macro (available upon request) to identify nuclear boundaries (UV channel) and speckles (Cy5.5 channel). Overlap and speckle counting was automated used the Biovoxxel toolbox (http://imagej.net/BioVoxxel_Toolbox) with the Speckle Inspector tool. Plots, confidence intervals, and significance calculations (two-tailed Kolmogorov–Smirnov test) were produced in Prism 7 (GraphPad). HeLa cells are listed in the database of commonly misidentified cell lines (maintained by ICLAC), but they are regularly used in for this assay (PLA) and in studying this interaction (YAP:TEAD), and both HeLa and 293T lines were obtained directly from the ATCC immediately prior to use (i.e. were not received from an affiliate laboratory). Mycoplasma testing took place at the ATCC. No authentication was performed beyond that done by the vendor.

**Data availability.** The data and computer scripts that support the findings of this study are available from the authors on reasonable request; data containing trade secret or proprietary information may not be provided. The SDGF and SDPR vectors are available upon request, pending a Materials Transfer Agreement with the FHCRC. The expression cassette sequences for the SDGF and SDPR vectors, which include the coding sequence for TB1G1, were deposited to GenBank under accession numbers MF958494 (SDGF) and MF958495 (SDPR).

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

## Acknowledgements

This work was funded by NIH Grants R01CA114567 (J.M.O.) and R01CA155360 (J.M. O.); A Washington Research Foundation Innovation Fellowship through the University of Washington Institute for Protein Design, and NIH Fellowship T32AG00005740 (ZRC); and Project Violet (www.projectviolet.org).

## Author contributions

Z.R.C., M.B., A.D.B, A.J.M, D.B., R.K.S., P.B., and J.M.O. conceived strategies. Z.R.C. and P.B. performed computational peptide design. Z.R.C., G.P.S., D.F., M.B., M.C., and M.G. designed and performed experiments, and analyzed data. Z.R.C. wrote the main manuscript. Z.R.C., D.F., R.K.S., and P.B. wrote the methods and prepared figures. All authors edited the manuscript.

## Additional information



