## [Peer Review File · Nature Communications]

Reviewers' comments:

Reviewer #1 (Remarks to the Author):

Olsen and coworkers provide in their manuscript an incredibly huge amount of data. They established a mammalian display system and displayed a set of 10,000 (!) cysteine-rich peptides and proteins on the cell surface. They analysed proteolytic susceptibility and display signals and thereby identified potentially stably folding candidates, of which hundreds (!) were expressed and the proteins were analysed by HPLC and other methods. Interestingly, these very impressive results are not discussed in the methods section and only indirectly mentioned in the abstract. Instead, the authors in the second section of the paper switch gears and provide data on a - from my point of view - very different topic, namely the generation of cysteine-rich libraries using their newly developed mammalian display platform and the high throughput FACS screening for TEAD binders thereby modulating the Hippo pathway. They successfully identified a candidate with subnanomolar KD that was able to interfere with TEAD:YAP binding. However, the paper lacks unity and in particular it lacks clarity. It is horrible to read and at some very important points it is impossible - at least to me - to understand the basic experimental concept. I did my very best to understand the concept behind the design of the library used for the screening of TEAD binders, but failed. Moreover, many details are missing that make it impossible to reproduce the experiments. As a consequence, despite the impressive data set I cannot recommend the paper for publication. At least the Results section has to be totally rewritten. The authors should also consider splitting their data into two manuscripts.

The most problematic points are detailed below:

- 1) Vector construction (online material): what is an improved CMV promoter? What is the sequence and assembly of the lentiviral LTRs? What is the DNA sequence context of the used cloning sites? Regarding the SDPR vector: what is the sequence of the C-terminal portion of the FasL domain with the adjacent (former) TEV cleavage sites?
- 2) It remains unclear why Phe Trp and Tyr besides Lys and Arg have been eliminated when using trypsin, that cleaves after basic residues as stated in the results section. The authors seem to hide the answer in line 596.
- 3) Line 102: The material source of the 10,000 peptides/proteins is not given and the experimental strategy, how these peptide and protein fragments were generated and how they were incorporated into the display vector is not provided. Have all 10,000 genes been ordered from a supplier for gene synthesis or where some of them PCR amplified from cDNA/chromosomal DNA (if yes what source)? Was the DNA sequence of each construct verified?
- 4) Line 102: What bioinformatics criteria were used to identify proteins to be included in the CDP library? What means in this context the selection criterium cysteine-rich?
- 5) Fig 1C: The nature of the construct is unclear. I presume it contains elafin, MAaCv or YAP fusion to C9 in place of the XCXXXCXX peptide indicated in Fig. 1A. This should be clarified
- 6) Line 124: How were these 604 library members produced? The statement at line 521 "were cloned into our secreted, soluble protein production vector" is not very useful. Not even the transfection host is given (vector name and host cells may be provided in ref 1,2).
- 7) Line 130: The statement "properly-folded soluble peptides (1-2 peaks) are more often found to fold well on the surface" is an interpretation. Better: , "properly-folded soluble peptides (1-2 peaks) are more often found to be protease resistant upon cell surface display"
- 8) Line 177: The authors used "a Rosetta protein design approach to identify an optide capable of

binding TEAD". Even with intensive reading the methods section (the term Rosetta is not mentioned there), it remains unclear what was done in general. If I understand it right, a de novo design strategy was applied. It remains unclear, why a 3-helix motif was chosen. Many stable cysteine-rich peptides (knottins etc) contain triple beta-sheets. The structure model of the resulting scaffold is hidden in Figure 4 D,G

9) It remains totally unclear to me, how the authors generate a library of optides based on the Rosetta design. Is this a mixture of sequence predictions from Rosetta? If yes what is the sequence diversity at which position? how many different sequences are represented in the library and how? Was an predicted 3-helix bundle scaffold generated by Rosetta that contained randomized amino acids at surface-exposed position? If yes which amino acids replacements were allowed at which position? Which random coding scheme was used, NNS codon mutagenesis or trinucleotide or has each library candidate been synthesized individually? Accordingly line 560: What is the nature of the pooled oligonucleotide library? What have the library members in common and What is different? Why is in the optide library in figure 3 the number of amino acids spacing the cysteines different from clone to clone? Obviously, latest at this point the reader gets completely lost.

10) The sequences of TB1G1 is hidden at the bottom of figure 4B, the TB1G2 sequence is not provided. An extra figure showing the proposed structure and the sequences of the binders would be useful.

11) Saturation mutagenesis Fig.4: It remains unclear how monovalent staining was achieved. What is meant with "Bottom row: monovalent 2-step streptavidin"? What is meant with warm and cool colors in 4B?

12) The legend in figure 4B does obviously not correspond to the figure. D, TB1G2 should read C, TB1G2: E, TB1G2-W40P should read D, TB1G2W40P.

13) SPR measurements depicted in Fig. 4: With a picomolar or single digit nanomolar KD one would expect a very low off-rate. However, fast dissociation is seen. This requires an explanation.

Reviewer #2 (Remarks to the Author):

The paper "Mammalian display screening of diverse cysteine-dense peptides for difficult to drug targets" by Crook et al. describes an approach to express cysteine-rich peptides, referred to as optides, and to these screen for their ability to interfere with protein-protein interactions. The authors go on to use the technique to isolate and characterize an optide capable of inhibiting interactions between YAP and TEAD in the Hippo signaling pathway. Cysteine-rich peptides have previously been identified as an exciting class of potentially therapeutic agents, and the authors explain and demonstrate that this class of reagents may be successfully developed against otherwise druggable targets.

The paper describes a broad range of experiments to characterize the screening system and the isolated molecules. The experimental work is elegant, competent, and convincing, and the paper is very well written.

The choice of mammalian cells for expression is due to the author's expectation that yeast and bacteria would be less well suited to express this class of peptides, but it is not clear if this expectation has been demonstrated in practice. As pointed out by the authors even their expression system only manages to express 17% of the clones as well folded protein.

The screening of cells is done using fluorescent streptavidin, but the authors should clarify that the target protein is biotinylated, both in the section "Mammalian surface display CDP screening.." and under "Mammalian surface display" in the On-line Methods section.

The peptides highlighted in green in figure 2B are very hard to see in the figure and may have to

marked more clearly. Also, although the HPLC traces corresponding to the green symbols are numbered in Figure 2D there is apparently no way to tell which trace pertains to which green symbol, or on which side of the dividing line in B they are located.

Reviewer #3 (Remarks to the Author):

Crook et al., describe a mammalian surface display system for cysteine rich peptides with good folding and solubility profiles. They have utilized this system to generate designed peptides that target YAP/TEAD interaction. The newly identified TEAD binder is further validated for its stability to protease, temperature and reducing conditions. Furthermore, the new molecule has been tested in cells and in competitive assays in presence of YAP. As mentioned by authors, the surface display method is routinely used to generate libraries for targeting protein-protein interactions. The process described here is a small improvement from the existing methods and makes use of the fact that mammalian systems express soluble peptides. Computational design of the peptides is an optional feature explored in the manuscript and adds more complexity to the design of new protein-protein inhibitors.

Overall, this is a technique-driven manuscript. The methods presented in this manuscript seem robust and have the potential to be generalized for targeting other protein-protein interactions beyond the Hippo pathway. The biological insights remain, unfortunately, somewhat limited, which might be of a concern for Nature Communications. Therefore, the manuscript might be more suitable for specialized journals such as Nature protocols or Nature Methods.

Specific Comments:

- (1) Figs. 3D and 3E show the modeled TB1G1 and TB2G1 peptides bound to TEAD but how the modeling was done is not described in the methods.
- (2) It is very confusing to use the name TB1G2-W40P but the actual mutation is P40W. Also, it is better to indicate where P40 is located in Figure 3D.
- (3) SPR data showed that the TB1G1 binds to TEAD strongly with K_d of 31nM, yet Co-IP showed that rather high concentration (10uM) of TB1G1 is required for disrupting the YAP-TEAD interaction. Any explanation for this observation ?
- (4) As interface 3 is bigger than interface 2 in the structure of YAP-TEAD (3KYS), what is the rationale to target interface 2 rather than interface 3 using peptide?
- (5) The two peptides TB1G1 and TB2G1 bind to the pocket of TEAD different from that occupied by the cyclic YAP peptide (Zhou et al. FESEB J. 2015). As the cyclic YAP peptide is shorter than TB1G1 and TB2G1, what are the advantages of these peptides compared to the cyclic YAP peptide ?

Reviewer #1 (Remarks to the Author):

Olsen and coworkers provide in their manuscript an incredibly huge amount of data. They established a mammalian display system and displayed a set of 10,000 (!) cysteine-rich peptides and proteins on the cell surface. They analysed proteolytic susceptibility and display signals and thereby identified potentially stably folding candidates, of which hundreds (!) were expressed and the proteins were analysed by HPLC and other methods. Interestingly, these very impressive results are not discussed in the methods section and only indirectly mentioned in the abstract. Instead, the authors in the second section of the paper switch gears and provide data on a - from my point of view - very different topic, namely the generation of cysteine-rich libraries using their newly developed mammalian display platform and the high throughput FACS screening for TEAD binders thereby modulating the Hippo pathway. They successfully identified a candidate with subnanomolar KD that was able to interfere with TEAD:YAP binding. However, the paper lacks unity and in particular it lacks clarity. It is horrible to read and at some very important points it is impossible - at least to me - to understand the basic experimental concept. I did my very best to understand the concept behind the design of the library used for the screening of TEAD binders, but failed. Moreover, many details are missing that make it impossible to reproduce the experiments. As a consequence, despite the impressive data set I cannot recommend the paper for publication. At least the Results section has to be totally rewritten. The authors should also consider splitting their data into two manuscripts.

Thank you for taking the time to make such detailed comments on our manuscript. Your concern about a lack of cohesion is well taken, and while the editor has expressed a desire to maintain the work as a single paper, the introduction and results of the revised manuscript have been modified in many ways, but two in specific response to your concern: the introduction has been modified to put an increased emphasis on the need for a TEAD-targeted therapeutic; and the transition from the platform-centric first part of the paper into the TEAD-centric second part has been expanded a bit and smoothed to make it less abrupt. In this way, a reader is less likely to be distracted by what could be viewed as a sudden change in focus.

The most problematic points are detailed below:

1) Vector construction (online material): what is an improved CMV promoter? What is the sequence and assembly of the lentiviral LTRs? What is the DNA sequence context of the used cloning sites? Regarding the SDPR vector: what is the sequence of the C-terminal portion of the FasL domain with the adjacent (former) TEV cleavage sites?

*In our resubmission, we have added a supplementary figure [**Supplementary Fig. 1**] that includes cloning site and flanking sequence information, including mutations that differentiate SDGF (our standard vector) and SDPR (our vector for assessing surface protein content and/or protease resistance). The expression cassettes have also been submitted to Genbank, with accession numbers MF958494 (SDGF) and MF958495 (SDPR); their release date is scheduled for Mar 1, 2018 or the date of this manuscript's publication, whichever comes first. The "improved CMV promoter" has been better identified (simian CMV, with citation) in the resubmission; thank you for catching that. Regarding the LTR sequences, there is nothing exceptional from any other standard, commercial SIN LTR lentivector, which is why we did not elaborate upon it. However, upon request, we can provide the full vector sequence.*

2) It remains unclear why Phe Trp and Tyr besides Lys and Arg have been eliminated when using trypsin, that cleaves after basic residues as stated in the results section. The authors seem to hide the answer in line 596.

*The reasoning for mutating aromatic residues is to allow for use of chymotrypsin in peptide protease resistance testing, as was used in **Fig. 6e** (formerly **5e**); this rationale has been better explained in the resubmission (**Results**, subsection “**Diverse native CDPs fold properly in mammalian display**”, paragraph 1; and **Methods**, subsection “**SDGF surface display vector construction and cloning.**”].*

3) Line 102: The material source of the 10,000 peptides/proteins is not given and the experimental strategy, how these peptide and protein fragments were generated and how they were incorporated into the display vector is not provided. Have all 10,000 genes been ordered from a supplier for gene synthesis or where some of them PCR amplified from cDNA/chromosomal DNA (if yes what source)? Was the DNA sequence of each construct verified?

*The **Methods** sub-section “**SDGF surface display vector construction and cloning**” identifies both Twist Bioscience and CustomArray Inc. as vendors of pooled oligonucleotides, and lists several strategies for cloning. PCR primers are listed in that same sub-section. However, it could have been worded more clearly; hence, a brief statement regarding the cloning strategy has been included in the relevant **Results** section [subsection “**Diverse native CDPs fold properly in mammalian display**”, paragraph 1], and the aforementioned methods sub-section has also been reworded for clarity, in the resubmission. DNA sequences of library members, for all pooled screens, were confirmed during high throughput sequencing, using the mapping software Bowtie2 as listed in the **Methods** subsection “**Next Generation Sequencing**”. However, the initial submission did not include scoring criteria, which has been added to this subsection in the resubmission. Notably, we were sure to eliminate any frameshift mutated sequences from analysis, which could otherwise score well by DNA mapping but would be partly or completely unaligned if more relevant but computationally taxing protein alignment mapping were applied.*

4) Line 102: What bioinformatics criteria were used to identify proteins to be included in the CDP library? What means in this context the selection criterium cysteine-rich?

*Information regarding the selection criteria for the high diversity CDP library has been added to the **Methods**, subsection “**High Diversity Native CDP Library Selection**”. In brief, the January 2014 UniProt database was filtered for proteins containing segments of 30-50 amino acids that contain 6, 8, or 10 cysteines. They were further filtered to eliminate known CDC and FDA toxins for safety reasons, resulting in ~96,000 peptides. A taxonomy-weighted random selection was then applied for selection of 9,999 peptides.*

5) Fig 1C: The nature of the construct is unclear. I presume it contains elafin, MAaCv or YAP fusion to C9 in place of the XCXXXCXX peptide indicated in Fig. 1A. This should be clarified

We can understand this view, and as it does not add a significant scientific or stylistic element, it has been removed.

6) Line 124: How were these 604 library members produced? The statement at line 521 “were cloned into our secreted, soluble protein production vector” is not very useful. Not even the transfection host is given (vector name and host cells may be provided in ref 1,2).

*In our resubmission, we have provided a better summary on the soluble peptide production methodology in the **Methods** subsection “**CDP Production and Purification.**” It still references the articles that contain the full protocol, rather than cut-and-paste a published protocol into our methods section, but we agree that there is a happy medium to be found between “just referencing the full protocols elsewhere” and “full copy-pasted protocols in our publication”. We hope that our resubmission achieves this.*

7) Line 130: The statement “properly-folded soluble peptides (1-2 peaks) are more often found to fold well on the surface” is an interpretation. Better: , “properly-folded soluble peptides (1-2 peaks) are more often found to be protease resistant upon cell surface display”

The phrasing of the “properly folded peptides” statement is correct as written, but a qualifier defining “properly folded” as high content / trypsin resistant was added to ensure that a reader sees “properly folded” in the context of our surface display protein content measurement. It would not be appropriate to label those peptides as “protease sensitive”, because our determination of proper surface folding incorporates both protease resistance and surface expression (a.k.a. protein content).

8) Line 177: The authors used “a Rosetta protein design approach to identify an optide capable of binding TEAD”. Even with intensive reading the methods section (the term Rosetta is not mentioned there), it remains unclear what was done in general. If I understand it right, a de novo design strategy was applied. It remains unclear, why a 3-helix motif was chosen. Many stable cysteine-rich peptides (knottins etc) contain triple beta-sheets. The structure model of the resulting scaffold is hidden in Figure 4 D,G

See below for the answers to points 8 and 9.

9) It remains totally unclear to me, how the authors generate a library of optides based on the Rosetta design. Is this a mixture of sequence predictions from Rosetta? If yes what is the sequence diversity at which position? how many different sequences are represented in the library and how? Was an predicted 3-helix bundle scaffold generated by Rosetta that contained randomized amino acids at surface-exposed position? If yes which amino acids replacements were allowed at which position? Which random coding scheme was used, NNS codon mutagenesis or trinucleotide or has each library candidate been synthesized individually? Accordingly line 560: What is the nature of the pooled oligonucleotide library? What have the library members in common and What is different? Why is in the optide library in figure 3 the number of amino acids spacing the cysteines different from clone to clone? Obviously, latest at this point the reader gets completely lost.

*The Rosetta design methodology is in the two sub-sections of **Methods**, formerly titled “**TEAD-Binding Optide Library Computational Design: Scaffold Construction.**” and “**TEAD-Binding Optide Library Computational Design: Interface Design**”, but it was an oversight to not include the word “Rosetta” in the sub-section titles, so this was added in the resubmission. A distilled description of the methodology was also added to the **Results** subsection “**Mammalian CDP screening to identify TEAD-binding optides**” paragraph 2, to provide more clarity. There was no general algorithm as to which positions were mutated, or what amino acids were allowed; it is entirely dependent on the particular design. Mutagenesis was similarly achieved by the design algorithm, rather than random (e.g. NNS). As to why 3-helix scaffolds were chosen, the portions of the YAP-TEAD interaction most crucial to functional binding are interfaces 2 and 3, both of which are helical in nature on YAP. Scaffolds including both sheets and helices are indeed part of many design libraries, but we found sufficient diversity within helix-rich scaffolds. Native knottins, such as the 3-sheet motif you refer to, are indeed quite stable. However, state-of-the-art Rosetta methodology performs poorly with scaffolds that resemble native knottins, as most native knottins are rich in loops devoid of secondary structure and primarily stabilized by disulfides. Rosetta performs best with scaffolds that have little-to-no unstructured regions. Hence, we took inspiration from knottins and knottin-like peptides in the size of the peptides in the library (30-41 amino acids) and the number of disulfides (3), but chose secondary structures and disulfide topology optimized for potential success at designing a YAP-inhibiting CDP. More clarity on these design choices has been included in the revised manuscript, **Results** subsection “**Mammalian CDP screening to identify TEAD-binding optides**” paragraph 3.*

10) The sequences of TB1G1 is hidden at the bottom of figure 4B, the TB1G2 sequence is not provided. An extra figure showing the proposed structure and the sequences of the bidners would be useful.

*The sequences of TB1G1 and TB1G2 are found in the **Supplementary Table 4** (formerly 3). However, we agree that they could be more prominently visible, so in our resubmission they have been included in **Fig. 4h** (formerly 3h) (TB1G1), **Fig. 5c,d** (formerly 4c,d) (TB1G2 and TB1G2-W40P), and **Supplementary Fig. 3** (TB2G1).*

11) Saturation mutagenesis Fig.4: It remains unclear how monovalent staining was achieved. What is meant with “Bottom row: monovalent 2-step streptavidin”? What is ment with warm and cool colors in 4B?

*The description of the staining methodology demonstrated in **Fig. 5a** (formerly 4a), including “monovalent 2-step streptavidin”, has been altered for clarity in the resubmission, seen in **Results** subsection “**Platform flexibility facilitates rapid affinity maturation**” paragraph 1. The meaning of the color choice, warm vs. cool, is shown on the scale to the right (enrichment scores; high being warm, low being cold). Description of the calculation of enrichment score is found within the figure legend.*

12) The legend in figure 4B does obviously not correspond to the figure. D, TB1G2 should read C, TB1G2: E, TB1G2-W40P should read D, TB1G2W40P.

*Regarding the Figure 4 legend, thank you for catching the error in the K_D assignment to figures 5c and 5d (formerly 4c and 4d, mistakenly attributed to 4d and 4e, a leftover from a previous draft that was not caught). Regarding the **Fig. 5b** (formerly 4b) legend, we are unclear as to why you believe this is a mistaken match of description to figure. A heat map is relatively commonplace for saturation mutagenesis data presentation. We can understand why having the heat map and the color-coded structures within panel B, and not split into two panels, makes the figure legend somewhat lengthy for that panel. However, we feel that there is a stylistic benefit of having the enrichment score and color-coded amino acid ID as part of the heat map. It allows a reader to match a column’s enrichment behavior (via average color) to the color choice of that amino acid for the structures below. This provides a direct match between the SSM data and the structural context of a particular mutation or set of mutations, from within a single panel. That said, as this confused you, we clearly can make this a bit easier. Therefore, we have included asterisks within the heat map for the point mutations selected for incorporation into TB1G2, and have color-coded the text on the structures to make the match up easier for residues of particular interest. The legend has also been altered for clarity, as space allows.*

13) SPR measurements depicted in Fig. 4: With a picomolar or single digit nanomolar K_D one would expect a very low off-rate. However, fast dissociation is seen. This requires an explanation.

For a 1:1 binding interaction, the dissociation constant is determined by the ratio of off-rate to on-rate. For a tight binding interaction, a fast off-rate is compensated by an even faster on-rate. TB1G2 indeed has a faster off rate than is typical of antibodies with similar binding constants, as expected given a much smaller buried surface area at the interface. This is compensated by an extremely fast on-rate ($2.2 \times 10^7 M^{-1}s^{-1}$). The values observed for the TEAD/ligand interactions are completely within binding regimens observed for known protein/protein interactions, and we feel that this requires no elaboration in the manuscript.

Reviewer #2 (Remarks to the Author):

The paper “Mammalian display screening of diverse cysteine-dense peptides for difficult to drug targets” by Crook et al. describes an approach to express cysteine-rich peptides, referred to as optides, and to these screen for their ability to interfere with protein-protein interactions. The authors go on to use the technique to isolate and characterize an optide capable of inhibiting interactions between YAP and TEAD in the Hippo signaling pathway. Cysteine-rich peptides have previously been identified as an exciting class of potentially therapeutic agents, and the authors explain and demonstrate that this class of reagents may be successfully developed against otherwise druggable targets.

The paper describes a broad range of experiments to characterize the screening system and the isolated molecules. The experimental work is elegant, competent, and convincing, and the paper is very well written.

The choice of mammalian cells for expression is due to the author’s expectation that yeast and bacteria would be less well suited to express this class of peptides, but it is not clear if this expectation has been demonstrated in practice.

*Thank you for your time in reviewing our manuscript, and thank you for the comments. We agree that we have not empirically demonstrated superiority of mammalian cells versus yeast or bacteria in displaying or screening diverse, native CDP libraries, which would have required substantial additional experiments and data processing to repeat the screens in two other organisms and platforms. The revised manuscript took an effort to ensure that this was not implied, and to state that this platform is meant to augment existing, well-established methods by facilitating the routine screening of challenging CDP scaffolds with a sensitive, adaptable mammalian platform (**Discussion**, paragraph 2). We also included a note that using the same cell line for screening and for soluble, endotoxin-free biologics production will reduce the risk that differences in species- or cell line-dependent post-translational modifications will result in a reduction in efficacy during the transition. As far as the possibility of yeast or bacteria performing similarly, it is difficult to speculate, because to our knowledge, a study of diverse native CDP space has not been carried out in either of the other two models mentioned. We do know that there are two kinds of CDP or CDP-like peptides that are routinely screened in bacteria / yeast: peptides based on well-characterized, native knottin scaffolds (of which there are fewer than 10 in regular use), and peptide variants of de novo scaffolds that are pre-evaluated by computational design for predicted stability. Neither of these strategies explore the structural diversity found in nature, and given the widespread convergent evolution and conservation of CDPs, we believe it is of great potential utility to explore this space using a platform most likely to be compatible with such diversity. In developing such a platform, the paucity of native proteins containing cysteine-rich domains in the bacterial and yeast secretomes drove us to mammalian cells.*

As pointed out by the authors even their expression system only manages to express 17% of the clones as well folded protein.

*To your point about the 17% success rate, and what that says about our overall theme of the utility of mammalian cells in such diverse screening, it is worth remembering that this represents all native CDPs and cysteine-rich protein fragments, regardless of protein context or any other characteristic other than those that led to inclusion in our library. This is an important point, leading us to move the figures demonstrating it from the supplement (formerly Supplemental Fig. 3) to the main manuscript (**Fig. 3**). There we see that if we focus on peptides whose context is similar to those routinely studied in yeast and bacterial models (CDPs that make up the majority or entirety of the protein product), the success rate increases to 40%. This further increases to 47% if we limit ourselves to annotated knottins and defensins, well-studied for stability and commonly used in yeast screens, but still largely annotated based on sequence homology and not functional validation of peptide stability. (The latter statistic is not stated in the manuscript, but can be calculated from the*

*data found in **Supplemental Table 3** [formerly 2]). It's difficult to know how favorably this compares to what one would expect in other models, given that a) such a study with diverse native peptides in other models has not been published, and b) smaller scale publications likely omit data on peptides that fail to demonstrate stability, making it difficult to assemble meta-data on other models' relative success rates. The closest study (which is still not very close), exploring protease sensitivity-based stability of diverse peptides, uses cysteine-free, computationally designed peptides in yeast display (Rocklin et al., Science 2017). Even in this setting, with the benefit of discrete protein context and powerful thermodynamic modeling, multiple rounds of computational design are required before the majority of designs demonstrate favorable stability. This is not to imply mammalian cells would be superior in their study, but is simply a reminder that peptide stability, even in the best of circumstances, is far from a given. We anticipate that similar iteration of our native CDP libraries will produce significant increases in the likelihood of folding success, and are in the early stages of such iteration.*

The screening of cells is done using fluorescent streptavidin, but the authors should clarify that the target protein is biotinylated, both in the section "Mammalian surface display CDP screening.." and under "Mammalian surface display" in the On-line Methods section.

Throughout the revised manuscript, we have made it clearer that our tested proteins are biotinylated.

The peptides highlighted in green in figure 2B are very hard to see in the figure and may have to be marked more clearly. Also, although the HPLC traces corresponding to the green symbols are numbered in Figure 2D there is apparently no way to tell which trace pertains to which green symbol, or on which side of the dividing line in B they are located.

*We have improved the clarity of **Fig. 2**, making it easier to identify the peptides that were selected for representative HPLC data. This was done by adding a panel to the HPLC data section representing a miniaturized Protein Content vs. Trypsin Resistance plot containing only those peptides selected for HPLC data demonstration.*

Reviewer #3 (Remarks to the Author):

Crook et al., describe a mammalian surface display system for cysteine rich peptides with good folding and solubility profiles. They have utilized this system to generate designed peptides that target YAP/TEAD interaction. The newly identified TEAD binder is further validated for its stability to protease, temperature and reducing conditions. Furthermore, the new molecule has been tested in cells and in competitive assays in presence of YAP. As mentioned by authors, the surface display method is routinely used to generate libraries for targeting protein-protein interactions. The process described here is a small improvement from the existing methods and makes use of the fact that mammalian systems express soluble peptides. Computational design of the peptides is an optional feature explored in the manuscript and adds more complexity to the design of new protein-protein inhibitors.

Overall, this is a technique-driven manuscript. The methods presented in this manuscript seem robust and have the potential to be generalized for targeting other protein-protein interactions beyond the Hippo pathway. The biological insights remain, unfortunately, somewhat limited, which might be of a concern for Nature Communications. Therefore, the manuscript might be more suitable for specialized journals such as Nature protocols or Nature Methods.

Specific Comments:

(1) Figs. 3D and 3E show the modeled TB1G1 and TB2G1 peptides bound to TEAD but how the modeling was done is not described in the methods.

*Thank you for your time in reviewing our manuscript, and thank you for the comments. We agree that we could have done better in explaining how the modeling (shown in **Fig. 4**, formerly **3**) was done. The process of Rosetta grafting and design (detailed in the **Methods** subsection “**TEAD-Binding Optide Library Rosetta Computational Design: Interface Design**”, and in the revision elaborated upon in the **Results** subsection “**Mammalian CDP screening to identify TEAD-binding optides**” paragraph 2) produces a model, in the style of a co-crystal structure, containing the designed peptide fit against the target protein (TEAD) in its most thermodynamically favorable conformation. The software exports this model in 3D structure compatible pdb format. This output, as processed in PyMol, was the basis for **Fig. 4d-e** and **Fig. 5b** (formerly 3d-e and 4b). The above **Methods** subsection was modified to make this more clear.*

(2) It is very confusing to use the name TB1G2-W40P but the actual mutation is P40W. Also, it is better to indicate where P40 is located in Figure 3D.

*We agree that this is somewhat confusing. Given that the second-generation TEAD binder (TB1G2) contains 5 mutations from first-generation TB1G1, while TB1G2-W40P contains 4 mutations (all but P40W), we looked at one of two options for nomenclature: either refer to the second-generation TEAD binder by naming all five mutations, which would require either substantial wordiness or confusing abbreviations, but would allow for the partner reversion variant to simply omit mention of P40W; or define the second generation binder by a new name once, but now base the reversion variant on this nomenclature (TB1G2-W40P). Given that the reversion mutant is only used in a limited number of experiments compared to TB1G2, we felt the second option was favorable. That said, we have tried to improve the clarity of this variant’s name, as shown in the revised **Results** subsection “**Platform flexibility facilitates rapid affinity maturation**” paragraphs 2 and 3.*

(3) SPR data showed that the TB1G1 binds to TEAD strongly with Kd of 31nM, yet Co-IP showed that rather high concentration (10uM) of TB1G1 is required for disrupting the YAP-TEAD interaction. Any explanation for this observation ?

To this point, we note that the relative concentrations of YAP and TEAD were not optimized for detailed kinetic analysis in the Co-IP assay, which is why we did not calculate IC₅₀ values. It seems likely that the relative concentrations are substantially above the K_D so that stoichiometric inhibition is playing a substantial role in this particular assay. However, for the orthogonal binding inhibition experiments in the surface display platform shown in **Fig. 7a-g** (formerly 6a-g), the concentration of TEAD is more tightly controlled and is well below that of the observed K_D of TB1G1-TEAD. In the context of this experiment, the half-maximal TB1G1 inhibition of YAP-TEAD binding is ~30 nM (again, not specifically measured because surface-bound YAP concentration is not well defined, making discrete IC₅₀ measurement of limited utility). This observed half-maximal inhibition is very close to the 31 nM binding constant of TB1G1-TEAD. We are reluctant to belabor this subtle difference in the manuscript, as it is somewhat of a distraction to the themes, but in our resubmission we made sure that the interpretation of the Co-IP data is not taken further than “TB1G1 inhibits YAP-TEAD binding in a dose-dependent fashion” so as to reduce confusion from this apparent dichotomy.

(4) As interface 3 is bigger than interface 2 in the structure of YAP-TEAD (3KYS), what is the rationale to target interface 2 rather than interface 3 using peptide?

We did, in fact, include designs against Interface 3 in the library; the revised **Methods** subsection “**TEAD-Binding Optide Library Rosetta Computational Design: Interface Design**” made this fact clearer in our description of the backbone residue segments selected for design superposition, assigning the segments to their respective interfaces. However, our only two binders happened to target Interface 2. We hypothesize that this is because Interface 2 is a relatively simple helix-in-a-groove, while Interface 3 involves a short helix-turn-helix. Rosetta scaffold design rarely incorporates short helix-turn-helix motifs, favoring longer secondary structure domains, so it was not terribly surprising that we had more success targeting Interface 2.

(5) The two peptides TB1G1 and TB2G1 bind to the pocket of TEAD different from that occupied by the cyclic YAP peptide (Zhou et al. FESEB J. 2015). As the cyclic YAP peptide is shorter than TB1G1 and TB2G1, what are the advantages of these peptides compared to the cyclic YAP peptide ?

In reference to the cyclic YAP-based peptide in Zhou et al., their methodology has certain advantages over ours. Being based on a simple YAP-based design, it requires no computational modeling, which is not immediately accessible to everyone. A shorter peptide is also indeed often easier to synthesize. However, we note that their reported affinity (0.29 μM with their best performer) was nearly three orders of magnitude weaker than that of TB1G2 (< 0.4 nM). Also, they did not demonstrate reduction resistance or intracellular activity of their top performing peptides, while TB1G2 is highly resistant to intracellular reducing conditions and effective at reducing intranuclear YAP:TEAD dimerization. Furthermore, their strategy required incorporation of non-natural amino acids to achieve their highest affinities. This is not an impediment when dealing with a discrete number of short, easy-to-synthesize peptides, but with regards to our goal of facilitating high diversity CDP screening, the incorporation of non-natural amino acids is not a realistic feature of high diversity peptide libraries. One would have to employ complicated genetic schemes that employ stop codon readthrough or other tricks that could limit cell viability. This is a valid discussion, and while we feel it would be an unnecessary distraction to belabor it in the manuscript to the depth done here, reference to this approach has been added to the revised manuscript in the **Results** subsection “**Mammalian CDP screening to identify TEAD-binding optides**”, paragraph 1.

REVIEWERS' COMMENTS:

Reviewer #1 (Remarks to the Author):

This is a resubmitted manuscript. My major concern with the first draft was lacking clarity. In this respect, the manuscript improved dramatically. It is now definitely worth being published in Nature Communications. The authors addressed all major concerns of the reviewers appropriately.

There is only one remaining minor point left:

In the results section, it is not stated in the text that a site saturation mutagenesis was done for affinity maturation of TBG1. This can only be deduced from the title of figure 5 and from the heat map in 5b. For further improved readability of the paper, it would be nice to provide this information e.g. at line 259 of the results section.

Reviewer #2 (Remarks to the Author):

The authors have satisfactorily addressed each of the points raised in my review, and I believe the paper will be of great interest to its readers.

Reviewer #3 (Remarks to the Author):

The authors have adequately addressed the points raised by all three reviewers and the manuscript now is acceptable for publication.

Reviewer #1 (Remarks to the Author):

This is a resubmitted manuscript. My major concern with the first draft was lacking clarity. In this respect, the manuscript improved dramatically. It is now definitely worth being published in Nature Communications. The authors addressed all major concerns of the reviewers appropriately.

There is only one remaining minor point left:

In the results section, it is not stated in the text that a site saturation mutagenesis was done for affinity maturation of TBG1. This can only be deduced from the title of figure 5 and from the heat map in 5b. For further improved readability of the paper, it would be nice to provide this information e.g. at line 259 of the results section.

*A statement for the use of site saturation mutagenesis for affinity maturation is found in the above-referenced section of the **Results** (Subsection "**Platform flexibility facilitates rapid affinity maturation**", second paragraph, [formerly] first sentence). However, it is stated in the second part of an unnecessarily long compound sentence, which may explain why it was missed. In the current draft, that sentence has been split into two sentences, reading as such:*

"For affinity maturation of TB1G1, we used a monovalent, two-step incubation with 20 nM biotinylated TEAD and streptavidin-Alexa Fluor 647. Variation was achieved by site saturation mutagenesis, making a library of every possible non-cysteine substitution."

This minor change will hopefully make it somewhat easier for readers to identify the technique used without feeling the need to reference figure legends / methods.

Reviewer #2 (Remarks to the Author):

The authors have satisfactorily addressed each of the points raised in my review, and I believe the paper will be of great interest to its readers.

Reviewer #3 (Remarks to the Author):

The authors have adequately addressed the points raised by all three reviewers and the manuscript now is acceptable for publication.